# Invariant γδTCR natural killer-like effector T cells in the naked mole-rat

Guillem Sanchez Sanchez [1,2,3,4], Stephan Emmrich [5], Maria Georga [6], Ariadni Papadaki [6], Sofia Kossida [6], Andrei Seluanov [5,7], Vera Gorbunova [5,7] & David Vermijlen [1,2,3,4] ✉

The naked mole-rat (*Heterocephalus glaber*) is a long-lived rodent species showing resistance to the development of cancer. Although naked mole-rats have been reported to lack natural killer (NK) cells, γδ T cell-based immunity has been suggested in this species, which could represent an important arm of the immune system for antitumor responses. Here, we investigate the biology of these unconventional T cells in peripheral tissues (blood, spleen) and thymus of the naked mole-rat at different ages by TCR repertoire profiling and single-cell gene expression analysis. Using our own TCR annotation in the naked mole-rat genome, we report that the γδ TCR repertoire is dominated by a public invariant Vγ4-2/Vδ1-4 TCR, containing the complementary-determining-region-3 (CDR3)γ CTYWDSNYAKKLF / CDR3δ CALWELRTGGITAQLVF that are likely generated by short-homology-repeat-driven DNA rearrangements. This invariant TCR is specifically found in γδ T cells expressing genes associated with NK cytotoxicity and is generated in both the thoracic and cervical thymus of the naked mole-rat until adult life. Our results indicate that invariant Vγ4-2/Vδ1-4 NK-like effector T cells in the naked mole-rat can contribute to tumor immunosurveillance by γδ TCR-mediated recognition of a common molecular signal.

The naked mole-rat (*Heterocephalus glaber*) is an eusocial mammal, highly specialized for life underground through sensory and ecophysiological adaptations, but it has also been used in biomedical studies to investigate for example pain, hypoxia, cardiac function, and cancer[1–3]. Naked mole-rats are highly resistant to cancer: thousands of individual animals have been monitored over decades in biomedical research laboratories, as well as in zoo colonies, and only six cases of tumors have been found[1]. Cell-intrinsic mechanisms, for example driven by the production of high molecular mass hyaluronan, can contribute to this resistance against cancer cell development[1,2]. However, it has been suggested more recently that cell-extrinsic factors such as immune cells can play major roles in the naked mole-rat cancer

resistance[3–5]. It is therefore intriguing that the naked mole-rat lack canonical natural killer (NK) cells, a lymphocyte population with potent anti-tumor activities[6]. In contrast, the naked mole-rat has been shown to possess large ectopic cervical thymi in addition to the canonical thoracic thymus, organs where T cells are generated[7]. Furthermore, naked mole-rat thymi do not show involution upon aging and there are indications that the naked mole-rat T cell compartment shows a bias towards γδ T cells[7].

γδ T cells are the "third" type of lymphocytes, besides αβ T cells and B cells, that can rearrange genes at the DNA level in order to generate variable antigen receptors[8,9]. These three cell lineages have been conserved seemingly since the emergence of jawed vertebrates,

[1]Department of Pharmacotherapy and Pharmaceutics, Université Libre de Bruxelles (ULB), Brussels, Belgium. [2]Institute for Medical Immunology, Université Libre de Bruxelles (ULB), Gosselies, Belgium. [3]ULB Center for Research in Immunology (U-CRI), Université Libre de Bruxelles (ULB), Gosselies, Belgium. [4]WELBIO Department, WEL Research Institute, Wavre, Belgium. [5]Department of Biology, University of Rochester, Rochester, NY, USA. [6]IMGT®, the international ImMunoGenetics information system®, Institut de Génétique Humaine (IGH), Centre National de la Recherche Scientifique (CNRS), Université de Montpellier (UM), Montpellier, France. [7]Department of Medicine, University of Rochester Medical Center and Medicine, University of Rochester, Rochester, NY, USA. ✉e-mail: David.Vermijlen@ulb.be

with the notable exception of squamate reptiles that lost γδ T cells[10], while a similar tripartite subdivision exists even in jawless vertebrates, such as lamprey and hagfish[11,12]. The rearrangements of the TRG (gamma chain) and TRD (delta chain) loci take place in the thymus during the development of γδ T cells from a common αβ/γδ T cell precursor[13]. During TRG or TRD rearrangement, a V (variable), D (diversity; only for TRD), and J (joining) gene are joined to form a final T cell receptor (TCR) chain. The variability created during the V(D)J recombination is significantly enhanced by the junctional diversity, which includes the introduction of random nucleotides (N additions), a process mediated by the enzyme Terminal deoxynucleotidyl Transferase (TdT). The most variable domain, that includes these N additions and is usually accountable for antigen recognition, is found in the complementarity determining region 3 (CDR3) of the TRG and TRD chains. The pairing of a single TRG with a TRD chain at the protein level results in the final TCR expressed on the surface of the γδ T cell. Based on their TCR repertoire, mouse and human γδ T cells can be divided into innate subsets that possess all the same/similar invariant CDR3 and adaptive subsets that possess a variable TCR repertoire[8]. The innate γδ T cells are typically produced in fetal waves before birth[14,15]. Notwithstanding some examples of MHC-restricted γδ TCRs[16], the γδ TCR is usually neither limited by MHC restriction nor primarily focused on processed antigens[9]. Still, a general rule of ligand recognition by the γδ TCR is hampered by the lack of ligand knowledge for the majority of γδ TCRs[17–19]. However, based on the known γδ TCR ligands, it appears that they have the potential to recognize diverse, largely host-derived, molecular entities[17,18,20]. Of note, in recent years an emerging role for butyrophilin (BTN) and butyrophilin-like (BTNL) molecules became evident in the TCR-dependent regulation of γδ T cells, involving TRGV regions outside the CDR3[9,21].

γδ T cells can perform cancer immunosurveillance: for example, the mouse invariant Vγ5Vδ1 TCR senses cell normality in the skin epidermis[22,23] and the (semi)invariant human Vγ9Vδ2 TCR senses the increased production of phosphorylated metabolites (known as phosphoantigens) in cancer cells[24]. As a consequence, γδ T cells are now being explored as a viable and promising approach for cancer immunotherapy in human, with the Vγ9Vδ2 T cell subset as a main target[25–27].

Since γδ T cells are important in cancer immunosurveillance and thus could contribute to the natural cancer resistance observed in the naked mole-rat, we investigated the presence of these unconventional T cells in the naked mole-rat and characterized them in detail, comparing with conventional αβ T cells, in peripheral tissues (blood, spleen) and in both thymus types. This required the annotation of the four TR loci (TRA, TRB, TRG and TRD) in *Heterocephalus glaber*. We found a striking prevalence, at a wide age range, of a γδ T cell population that possess an invariant γδ TCR and an NK-like cytotoxic effector phenotype.

## Results

### Annotation of naked mole-rat (*Heterocephalus glaber*) TR loci
To determine the type of expressed naked mole-rat TCRs by high-throughput sequencing, we first annotated the TR loci (TRG, TRAD, and TRB) of *Heterocephalus glaber* (Fig. 1). While annotations were conducted for the TRG and TRB loci in both the maternal and paternal chromosomal assemblies, the annotation for the TRAD locus was exclusively carried out in the maternal assembly. This decision was prompted by the presence of gaps in the paternal assembly of the TRAD locus, indicative of lower quality sequencing in that genome region. Of note, there were no differences observed in the presence or order of TR genes between the annotated maternal and paternal haplotypes for TRB and TRG loci. The evaluation of the accuracy of the assemblies was performed through the detailed revision of the estimated number of variable, diversity, joining and constant genes at the locus, along with their

organization, in comparison with the data of the most closely related evolutionary species.

The TRG locus of the naked mole-rat is located on chromosome 1, spanning ~333 kilobases (kb) from 10 kb upstream of the most 5' gene, *TRGV1-1*, to 10 kb downstream of the most 3' gene, *TRGC5*. The delimitation of the locus was based on the investigation of the "IMGT bornes", which are coding genes, other than TR, that remain conserved across different species and are located upstream of the first or downstream of the last gene of the locus, respectively[28]. In both maternal and paternal assemblies, the 3' borne of TRG locus (*STARD3NL*) is detected 5 kb downstream of *TRGC5*. In the maternal assembly only, the 5' borne of TRG (*AMPH*) is present, 34 kb upstream the *TRGV1-1* (Fig. 1A). Within this locus 31 TRG genes were detected, consisting of 15 TRG variable genes (TRGV) (1 F, 3 ORF, 11 P), 11 TRG joining genes (TRGJ) (1 F, 6 ORF, 4 P), and 5 TRG constant genes (TRGC) (3 F, 2 P) (Fig. 1A). All the genes were present in both maternal and paternal assemblies and polymorphisms between the two assemblies were detected in 6 genes: 3 TRGV genes (*TRGV1-3*, *TRGV1-4* and *TRGVD*), 1 TRGJ gene (*TRGJ3-2*) and 2 TRGC genes (*TRGC2* and *TRGC5*). A gene is assigned ORF (open reading frame), rather than F (functional), when the coding region has an open reading frame without stop codons (like in a F) but shows (unlike the F designation) gene alterations in the recombination signals (compared to the canonical recombination signal sequences), deletions of more than 3 amino acids in the V-REGION and/or changes of conserved amino acids that might lead to incorrect protein folding. It is important to note that the annotation of a gene as ORF (versus F) is a dynamic process, particularly when applied to new species, where these changes may be unique and characteristic of a specific species. A gene is assigned P (pseudogene) when the coding region has stop codon(s) and/or frameshift mutation(s) and/or a mutation affecting the initiation codon. The naked mole-rat's TRG genes are organized into 5 V-J-C-CLUSTERs or "cassettes". The number of TRGV (1–6) and TRGJ (1–3) varies in each cluster, while all clusters contain one TRGC (Fig. 1A). This organization into different clusters or cassettes is also observed in the mouse TRG locus but not in the human TRG locus.

The TRB locus of the naked mole-rat is located on chromosome 22 and extends over a distance of 283 kilobases (kb), ranging from 10 kb upstream of the most 5' gene in the locus (*TRBV4*) to 10 kb downstream of the most 3' gene (*TRBV30*) (Fig. 1B). The 3' borne, *EPHB6*, is located 180 kb downstream of TRBV30. The TRB locus of naked mole-rat consists of 29 TRBV categorized into 26 TRBV subgroups (8 F, 8 ORF and 13 P in the maternal assembly and 9 F, 8 ORF and 12 P in the paternal assembly), 2 TRBD belonging to 2 TRBD sets (2 F), 13 TRBJ (6 F and 7 ORF) belonging to 2 TRBJ sets, and 2 TRBC (2 F) (Fig. 1B). The genes were present in both maternal and paternal assemblies, while polymorphic variations were observed in 5 genes across the two assemblies: 2 TRBV genes (*TRBV9* and *TRBV27*), 1 TRBJ gene (*TRBJ1-6*) and 2 TRBC genes (*TRBC1* and *TRBC2*).

The TRAD locus, thus containing the genes for both the TRA and TRD of αβ and γδ T cells respectively, of the naked mole-rat is located on chromosome 2 and spans 1368 kilobases (kb), ranging from 10 kb upstream of the most 5' gene (*TRAV1*), to 10 kb downstream of the most 3' gene (*TRAC*) (Fig. 1C). The 5' borne (*OR10G3*) and 3' borne (*DAD1*) have been identified 26 bp upstream of *TRAV1* and 20 kb downstream of *TRAC*, respectively (Fig. 1C). In the TRAD locus, 79 TRAV genes (39 F, 6 ORF and 34 P) were detected belonging to 39 TRAV subgroups. The naked mole-rat TRAD locus consists also of 11 TRDV genes (4 F, 2 ORF, and 5 P), 3 TRDD (2 F and 1 ORF), 4 TRDJ (1 F, 2 ORF, and 1 P), and 1 TRDC (F) (Fig. 1C). The TRDV genes are localized in 2 V-CLUSTERs: the 5' V-CLUSTER, in 5' of the TRDD, TRDJ, and TRDC, includes 10 TRDV genes and the 3' V-CLUSTER, downstream of the TRDC in inverted orientation, includes 1 TRDV. The TRDD, TRDJ, and TRDC genes are localized in 1 D-J-C-CLUSTER between the 5' V-CLUSTER and the 3' V-CLUSTER (Fig. 1C).

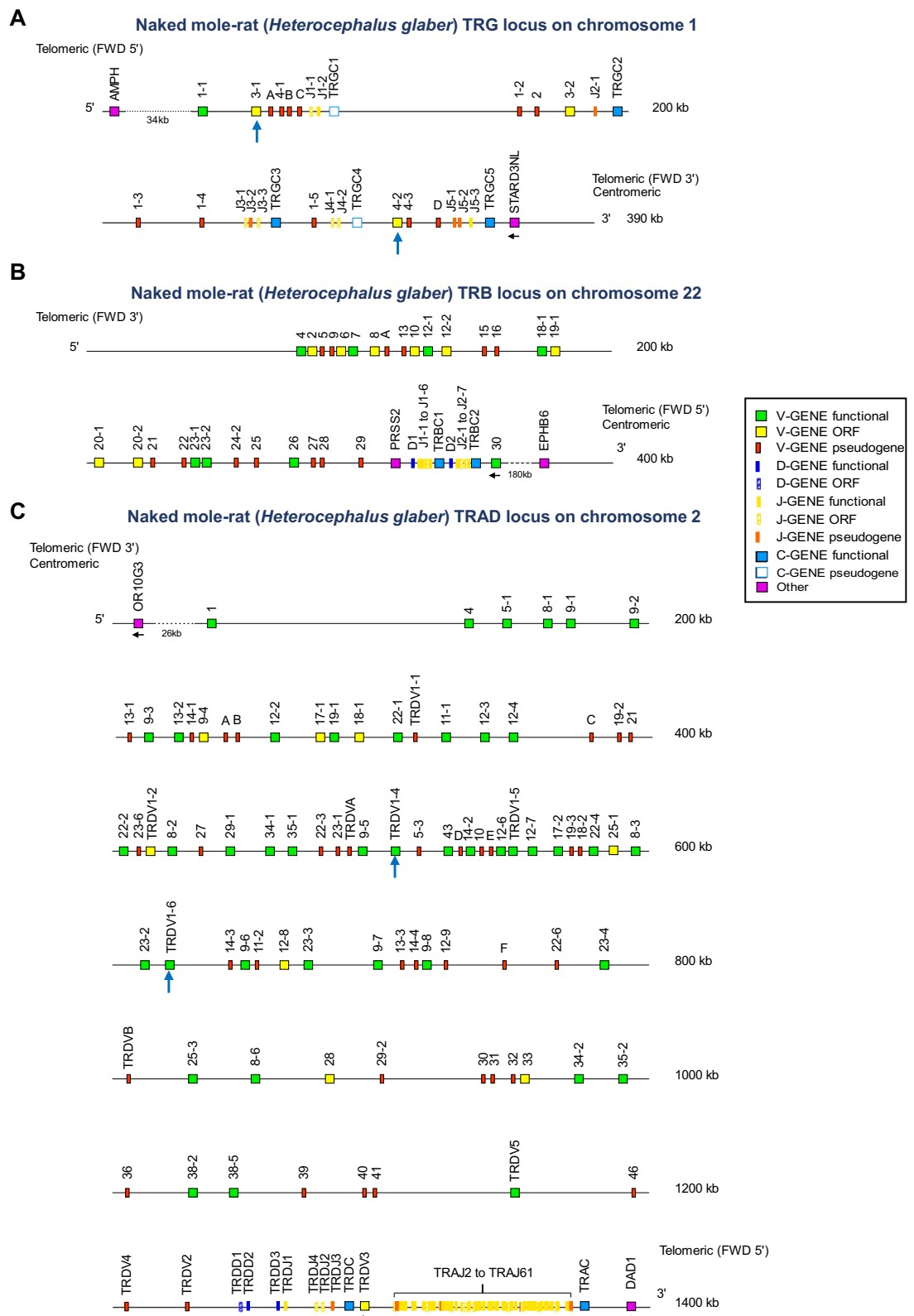

**A** Naked mole-rat (*Heterocephalus glaber*) TRG locus on chromosome 1

**B** Naked mole-rat (*Heterocephalus glaber*) TRB locus on chromosome 22

**C** Naked mole-rat (*Heterocephalus glaber*) TRAD locus on chromosome 2

## Comparison of naked mole-rat TRGV and TRDV genes with human and mouse

To gain a deeper understanding of the naked mole-rat's TR loci and their variable genes, an unrooted phylogenetic analysis was performed, employing a holistic approach that incorporated TRGV, TRDV, TRAV, and TRBV nucleotide sequences from *Heterocephalus glaber* along with the well-studied *Homo sapiens* and *Mus musculus* (Fig. 2; Supplementary Fig. 1). Within these trees, genes belonging to the same subgroup cluster together under the same branch, often with a corresponding human gene. The subgroup classification of the naked mole-rats' TR loci is primarily based on human subgroups, with a few exceptions such as *TRBV1*, *TRDV4*, and *TRDV5*. These exceptions arise from their significantly low genetic identity percentages, which hinder correspondence with human subgroups. In terms of TRGV subgroups,

**Fig. 1 | Locus representation of the *Heterocephalus glaber* TR loci of the assembly Naked mole-rat maternal (GCA_944319715.1). A** Naked mole-rat (*Heterocephalus glaber*) TRG locus on chromosome 1; the orientation of the TRG locus is forward (FWD). **B** Naked mole-rat (*Heterocephalus glaber*) TRB locus on chromosome 22; the orientation of the TRB locus is reverse (REV). **C** Naked mole-rat (*Heterocephalus glaber*) TRA/TRD locus on chromosome 2; the orientation of the TRA/TRD locus is reverse (REV). **A–C** The figures display the position, nomenclature and functionality of the TR genes according to the IMGT nomenclature[73]. The proposed nomenclature is considered provisional based on future data being available and involvement of IUIS at the appropriate time/level. The dotted line indicates the distance in kb between the locus and its IMGT bornes[28] (5' borne and the most 5' gene in the locus and/or 3' borne and the most 3' gene in the locus) on the top and bottom lines, respectively. These distances are not represented at scale and are not included in the numbers displayed at the right ends of these two lines. The telomeric and centromeric indications refer to the position of the locus in the chromosome. There are two telomeric ends referenced within the locus (one at the 5' and one at the 3' of the locus), with the centromeric indication added to its nearest end. Finally, the black arrows indicate an inverse transcriptional orientation in the locus while the blue arrows indicate the position in the TRG and TRD locus of V genes that are highlighted in the current study.

both the 11 variable genes of the naked mole-rat and the 13 variable genes of humans are classified in four subgroups. The most prevalent subgroup in both species is TRGV1, which consists of 5 variable genes in the naked-mole rat and 10 variable genes in humans (containing for example *TRGV8*). In mouse, no direct correspondence between its TRGV subgroups and human has been identified so far. However, this phylogenetic analysis reveals that subgroup TRGV1 (containing *Trgv1*, *Trgv2*, and *Trgv3*) and subgroup TRGV2 (containing *Trgv4*) of mouse are clustered together in the same branch with subgroup TRGV4 (containing the non-functional *TRGV11*) of human (Fig. 2A).

Interestingly, the naked mole-rat possess TRGV (*TRGV2*) and TRDV (*TRDV2*) genes that show high identity with the human *TRGV9* and *TRDV2* genes respectively (Fig. 2), while the mouse TRGV or TRDV genes do not show significant identity with these human genes. These human *TRGV9* and *TRDV2* genes compose the variable domains of the human semi-invariant Vγ9Vδ2 TCR expressed by the phosphoantigen-reactive γδ T cells, the most prevalent γδ T cell subset in human peripheral blood and which are regarded as innate T cells[8]. Of note, both the naked mole-rat *TRGV9* and *TRDV2* genes are pseudogenes (Fig. 1A, C).

In the mouse, *Trdv4* (often referred to as "Vδ1") is a main gene used in the invariant γδ TCR of mouse γδ T cells[8]. As for the human *TRDV2*, also for this "mouse invariant" TRDV gene a close orthologue exists in the naked mole-rat, *TRDV4* (Fig. 2B), but which is again annotated as a pseudogene (Fig. 1C). For the mouse TRGV genes used in invariant γδ TCRs (*Trgv5* and *Trgv6*), no clear orthologue exists in the naked mole-rat (Fig. 2A).

Thus overall, we identified naked mole-rat TRGV and TRDV genes showing high identity with mouse/human genes that are used in mouse and human invariant γδ TCRs. However, these naked mole-rat genes are pseudogenes, questioning the presence of conserved invariant naked mole-rat γδ TCRs.

### Selective conservation of BTN(L) members in the naked mole-rat

In recent years, is has become increasingly clear that members of the BTN(L) family are crucial for γδ T cell biology, especially for those γδ T cell subsets with (semi)invariant γδ TCRs[8]. Mouse Skint1/2 and Btnl1/Btnl6 are important for the development of skin Vγ5+ and intestinal Vγ7+ γδ T cells respectively[9,29–31], while human BTN3A1/BTN2A1 have been shown to be crucial for phosphoantigen-reactivity[32–35]. As naked mole-rat TRGV/TRDV genes with high identity to "invariant" human/mouse genes are pseudogenes (Fig. 2), we wondered whether the naked mole-rat lacks conservation of mouse/human BNT(L) members. Thus, we reconstructed an unrooted phylogenetic tree analysis of naked mole-rat BTN(L) amino acid sequences with their counterparts in mouse and human (Fig. 3). This showed that that some BTN(L) members were conserved across all three species (BTN1A1, BTNL9), while others were only shared by pairs of species (BTNL3/BTNL8 between human and naked mole-rat, BTNL2 between mouse and human and BTNL10 between mouse and naked mole-rat). Furthermore, members of the BTN3 family were specific to human while members related to Skint1 were specific to mouse species (Fig. 3). Thus, the naked mole-rat may possess a selected conservation of mouse/human BTN(L) members in its genome.

### The γδ TCR repertoire of the naked mole-rat is highly oligoclonal

To obtain an unbiased insight into the re-arranged and expressed TCR repertoire of naked mole-rat, we conducted TCR sequencing (TCR-seq) experiments targeting the CDR3 of TRA, TRB, TRG, and TRD chains on RNA derived from blood samples from 3 neonates, all from the same litter, and blood and spleen samples from 7 adults ranging from 3 to 11 years old, originating from 4 different litters. Analysis of general TCR repertoire properties revealed lower levels of N additions and shorter CDR3 in neonatal TRB, TRG, and TRD repertoires compared to adult repertoires (Fig. 4A, Supplementary Fig. 2A, B).

Interestingly, there was a huge bias, both in the neonatal and adult group, towards the usage of the *TRGV4-2* gene in the TRG repertoire (Fig. 4B, top panel, Supplementary Table 2). The neonatal group showed the strongest bias towards the *TRGV4-2* usage and thus the adult animals had a tendency to have more (though not statistically significant) expression of *TRGV1-1*, *TRGV3-1* and *TRGV3-2* compared to the neonatal group (Fig. 4B, top row). The TRD repertoire exhibited a strong preference for the utilization of *TRDV1-4* and *TRDV1-6* genes, with *TRDV1-4* being more prominently enriched in neonates and *TRDV1-6* in adults (Fig. 4B bottom row, Supplementary Table 2). Differences in junctional diversity, CDR3 nucleotide length and NDN size between neonates and adults had the tendency to be more prominent in TRDV1-6-containing than in TRDV4-1-containing sequences (Fig. 4C). While a similar strong overall bias in V gene usage could not be observed in the TRA and TRB repertoire, some differences were found between the neonatal and adult groups (Supplementary Fig. 2C, D).

Importantly, examination of repertoire clonality revealed massive enrichment in the TRG repertoire and an oligoclonal pattern in the TRD repertoire, with the most abundant clones dominating a significant portion of the total repertoire (Fig. 4D, E; Supplementary Fig. 3). The TRB repertoire exhibited high polyclonality, while the TRA repertoire only displayed oligoclonality in the neonatal group, albeit at lower levels compared to the TRG and TRD repertoires (Fig. 4D, E; Supplementary Fig. 3). Note that an increase in clonality, indicated by a higher frequency of clones with lower abundance in the total repertoire, was observed in the adult group from all repertoires (Fig. 4E), possibly related to a limited expression of TdT in naked mole-rats at birth. Consistent with reduced junctional diversity during the neonatal period in the naked mole-rat, the levels of TRB, TRG, and TRD repertoire sharing (as defined by the F value) within neonatal samples were higher compared to their adult counterparts (Fig. 4F). The higher TRD sharing was due to the enriched expression of TRDV1-4-containing CDR3 sequences (Fig. 4F bottom row). When performing neonate vs. adult paired comparisons, the neonatal group consistently showed higher sharing levels in all repertoires (Supplementary Fig. 4A).

### Invariant TRGV4-2 and TRDV1-4 CDR3 sequences dominate the naked mole-rat TRG and TRD repertoire

Because of the high bias in TRGV and TRDV usage in the naked mole-rat, we wondered whether this was associated with particular patterns at the level of TCR arrangement. Therefore, we conducted an analysis

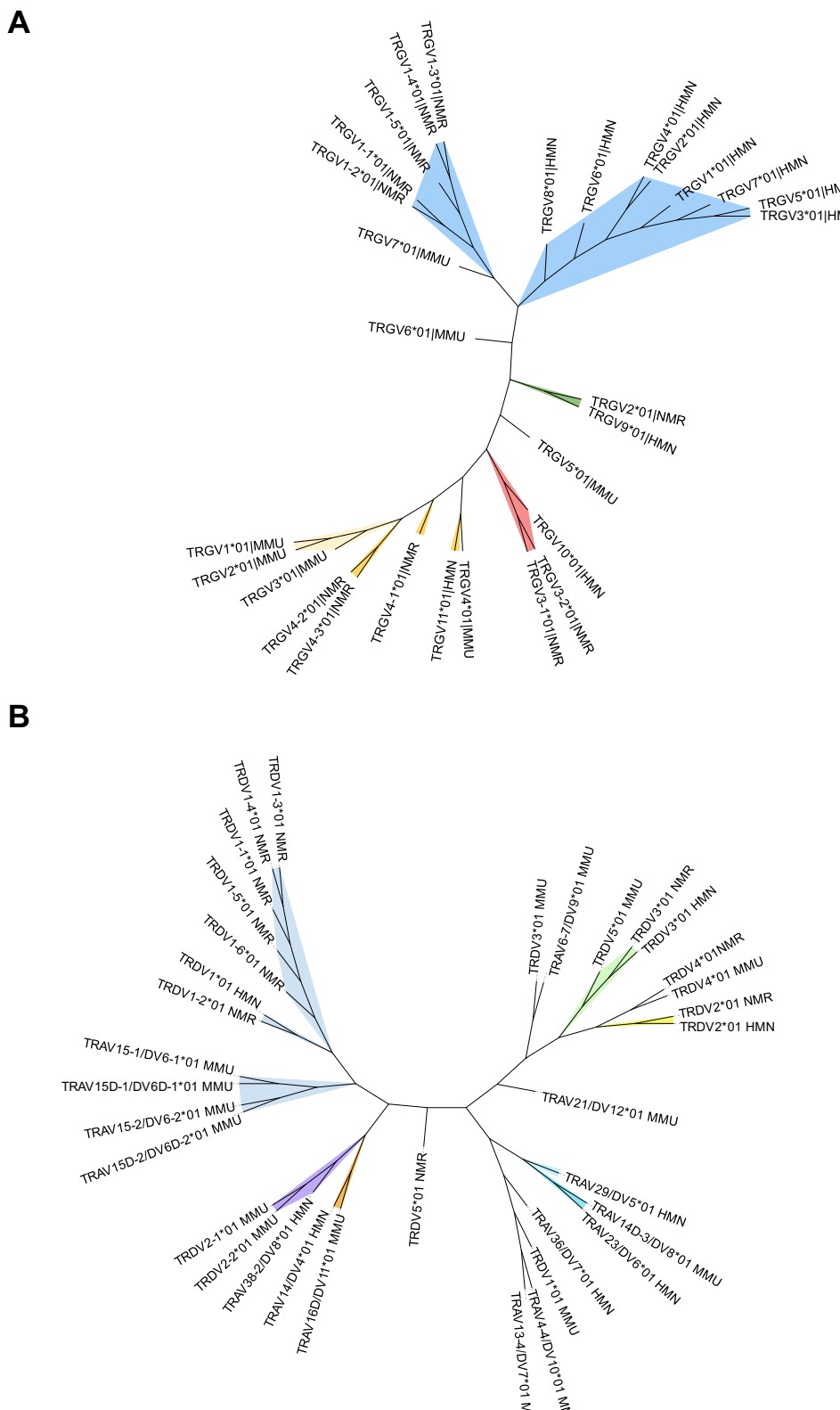

**Fig. 2 | Unrooted phylogenetic trees of TRGV (A) and TRDV (B) genes in *Homo sapiens, Mus musculus,* and *Heterocephalus glaber*.** The trees were constructed using the Maximum Likelihood method in MEGA software[79] The trees are based on nucleotide sequence alignments, generated with representatives of V-REGION sequences from each subgroup using MAFFT[80]. Visualization of the trees was carried out using iTOL v6[81], with distinct colors representing the different human subgroups and their orthologs. For instance, under (**A**) the TRGV1 subgroup includes the human *TRGV8* gene and the naked mole-rat *TRGV4-2* gene; under (**B**), the TRDV1 subgroup includes the human *TRDV1* gene and the naked mole-rat *TRDV1-4* gene. Species abbreviations are as follows: HMN *Homo sapiens*, MMU *Mus musculus*, NMR *Heterocephalus glaber*. Source data are provided as a Source Data file.

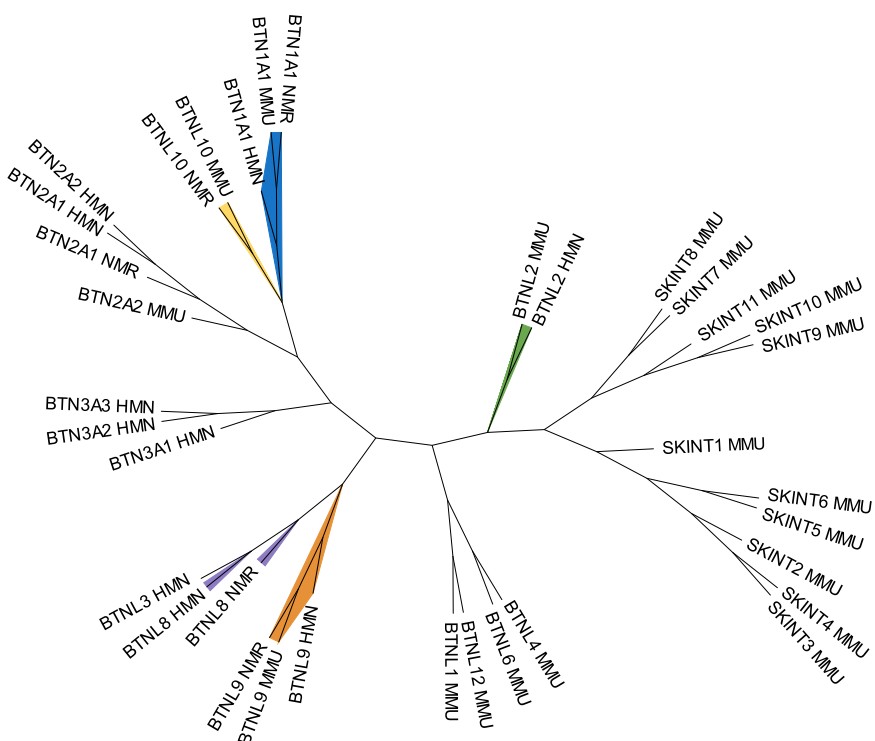

**Fig. 3 | Unrooted phylogenetic trees of BTN(L) molecules from *Homo sapiens*, *Mus musculus*, and *Heterocephalus glaber* species.** Maximum Likelihood tree of BTN(L) proteins was constructed utilizing MEGA software[79], based on amino acid MAFFT[80] sequence alignments of the BTN(L) and SKINT proteins obtained from NCBI. Protein sequences were used as there was enough information to reconstruct the phylogenetic tree. A single representative sequence from each protein family was selected for inclusion in the analysis. The tree was visualized with iTOL v6[81], and manually assigned colors indicate orthologous proteins, determined using the NCBI ortholog database. For example, the BTNL9 proteins from HMN, MMU, and NMR share a branch (colored orange) as orthologs. Species abbreviations are as follows: HMN *Homo sapiens*, MMU *Mus musculus*, NMR *Heterocephalus glaber*. Source data are provided as a Source Data file.

of CDR3 sequences. Remarkably, the TRG repertoire was overwhelmingly dominated by a highly prevalent public CDR3 sequence (that is: shared between different individuals), which was present in all the samples: CTYWDSNYAKKLF (Fig. 5A left panel; Supplementary Fig. 4B; Supplementary Fig. 5A).

This sequence, composed of a *TRGV4-2* gene and a *TRGJ5-3* gene, was germline-encoded without junctional diversity (Fig. 5B). Intriguingly, when aligning this sequence with invariant CDR3γ sequences from mice and humans[8], we found that the CTYWDSNYAKKLF sequence shared the same length and conserved amino acid intrinsic physicochemical properties as the human fetal-enriched invariant *TRGV8* CDR3 sequence[36] and the mouse invariant fetal-restricted *TRGV5* and *TRGV6* CDR3 sequence[37,38] (Fig. 5C).

Similarly, the TRDV1-4 repertoire exhibited high levels of public CDR3 sequences, accounting for more than 10% of the repertoire, while the TRDV1-6 repertoire displayed more modest values (Supplementary Fig. 4B). When comparing publicity between neonates and adults, the repertoire space occupied by these highly public sequences tended to be higher in neonates for TRDV1-4 and TRDV1-6 sequences, with a smaller difference observed in the case of TRG sequences (Supplementary Fig. 4B). The TRDV1-4 repertoire was dominated by a single CDR3 sequence: CALWELRTGGITAQLVF (Fig. 5A right panel; Supplementary Fig. 5B). This sequence consisted of *TRDV1-4*, *TRDD2*, *TRDD3*, and *TRDJ2* genes (Fig. 5D).

Both the highly prevalent invariant TRGV4-2/TRGJ5-3 sequence CTYWDSNYAKKLF and invariant TRDV1-4/TRDD2/TRDD3/TRDJ2 sequence CALWELRTGGITAQLVF contained short homology repeats (also known as microhomology regions)[8] that likely contribute to their formation (Fig. 5B, D). Finally, we observed a low-frequency public TRDV1-6-containing CDR3 sequence in distinct individuals

(Fig. 5A right panel), containing the same short homology repeats between TRDD2-TRDD3 and between TRDD3-TRDJ2 as the TRDV1-4 invariant sequence (Fig. 5D), but in contrast to the TRDV1-4 invariant sequence it does not possess a short homology repeat between the *TRDV1-6* and *TRDD2*, possibly explaining its lower frequency (Supplementary Fig. 6A, B).

In contrast to the TRG and TRD repertoire, no highly public sequences were detected in the TRA and TRB repertoire (Supplementary Fig. 4B; Supplementary Fig. 6C). However, a shared *TRAV1-TRAJ33* sequence was present at low levels among the most prevalent TRA sequences (Supplementary Fig. 6C). This TRA-TRAJ combination is highly prevalent among mucosal-associated invariant T (MAIT) cells, an innate-like T cell subset that is highly conserved across species and recognizes small non-peptidic molecules presented by MR1 molecules, a nonclassical MHC[39]. Alignment of the detected public TRAV1-TRAJ33 sequence from the naked mole-rat with the mouse invariant *Trav1-Traj33* MAIT sequence revealed a very high level of similarity (Supplementary Fig. 6D), suggesting that this sequence might be the TRA fingerprint of the conserved MAIT lineage, present at low frequencies (Supplementary Fig. 6E), in the naked mole-rat. Invariant natural killer T (iNKT) cells are another prototypical innate-like subset that have a semi-invariant TCR that is able to recognize glycolipid antigens presented by the MHC-like molecule CD1d[40]. Notably, no similar invariant rearrangements were detected in the naked mole-rat TCR ecosystem, which is in line with the absence of CD1d from their genome.

In sum, the naked mole-rat TRG and TRD repertoire is dominated by highly public invariant CDR3 sequences, likely directed by short-homology repeat recombination, while in the TRA repertoire a low-level MAIT-like sequence could be detected.

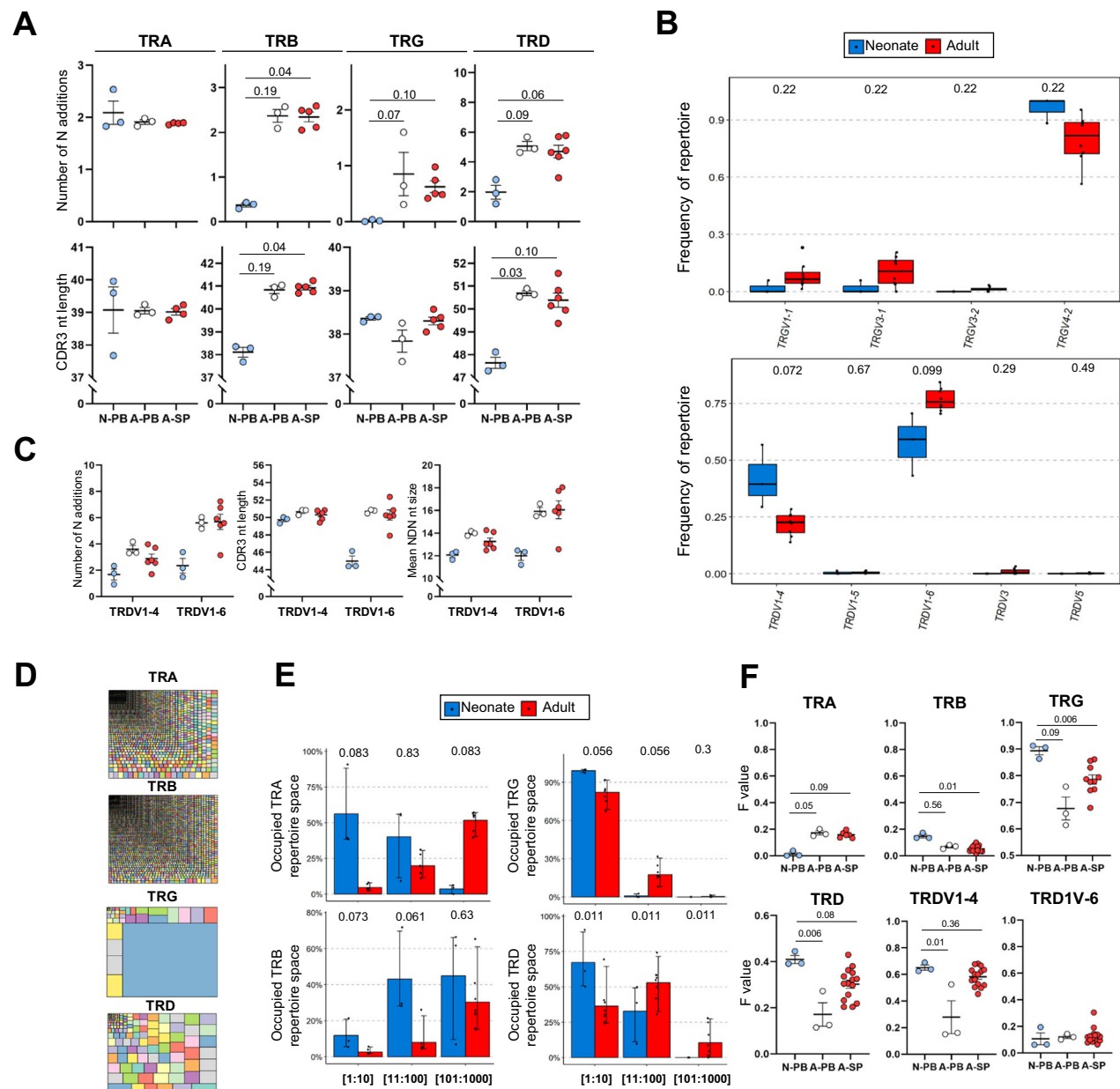

**Fig. 4 | The TCR repertoire expressed by the naked mole-rat is influenced by age and possesses an oligoclonal TCRγδ repertoire. A** Dot plots showing number of N additions (top) and CDR3 nucleotide length (bottom) in expressed CDR3α, β, γ, and δ sequences. Blue dots: neonate blood (N-PB, $n = 3$), white dots: adult blood (A-PB, $n = 3$), red dots: adult spleen (A-SP, $n = 4$–$6$). Each dot represents a naked mole-rat. Lines indicate means. **B** Boxplots of TRGV (top) and TRDV (bottom) usage of neonate blood samples ($n = 3$) and adult blood and spleen ($n = 7$). All detected V genes (expressed at RNA level) are displayed. The adult blood and spleen data sets were combined to increase statistical power. **C** Dot plots showing number of N additions (left), CDR3 nucleotide length (middle) and NDN nucleotide size (right) from TRDV1-4 and 6 repertoires. Blue dots: neonate blood, white dots: adult blood, red dots: adult spleen. Lines indicates means. **D** Example of a tree map visualization of a TRA, TRB, TRG, and TRD repertoires. **E** Clonality assessment of the distinct repertoires of neonate blood samples ($n = 3$) and adult blood and spleen ($n = 7$).

Clonotypes are grouped according to their abundance. 1:10 = first 10 clones, 11:100: 11–100th clone, 101:1000: 101–1000 clone. **F** Dot plots representing mean sharing values (*F* value) in the TRA, TRB, TRG, TRDV1-4, and TRDV1-6 repertoire of neonate (blood, blue dots (N-PB), $n = 3$) and adult animals (blood (A-PB), $n = 3$) and spleen (A-SP, $n = 4$–$6$). Repertoires of samples within an age group are compared pairwise and the mean level of sharing is plotted in the graph. Lines indicate means.
**B** Boxplots display the interquartile range (25–75th percentile) with the median line. Whiskers extend to the minimum and maximum values within a certain distance from the quartiles, with outliers shown as larger dots. **E** The error bars represent the 95% confidence interval around the mean value (height of the bar). **A**, **C**, **F** Data was analyzed using Kruskal-Wallis with Dunn's multiple test for pairwise comparisons. **B**, **E** Data was analyzed using Kruskal-Wallis with Holm correction for pairwise comparisons. Source data are provided as a Source Data file.

## γδ T cells from naked mole-rat diverge into two distinct transcriptomic profiles

To gain insights into the potential physiological role of γδ T cells and the associated significance of our observations regarding the γδ TCR, we employed TRUST4, a tool for extracting CDR3 sequences from single-cell sequencing data. We utilized a publicly available dataset

obtained from the blood of naked mole-rats, comprising samples from three young adult animals (around 3 years old) and three middle-aged animals (around 11 years old)[41]. After extracting CDR3 sequences and filtering for TCR-positive cells, six distinct clusters were identified that correlated well with previously identified clusters from the original study[41] (Fig. 6A, Supplementary Fig. 7A, B). The TRG and TRD

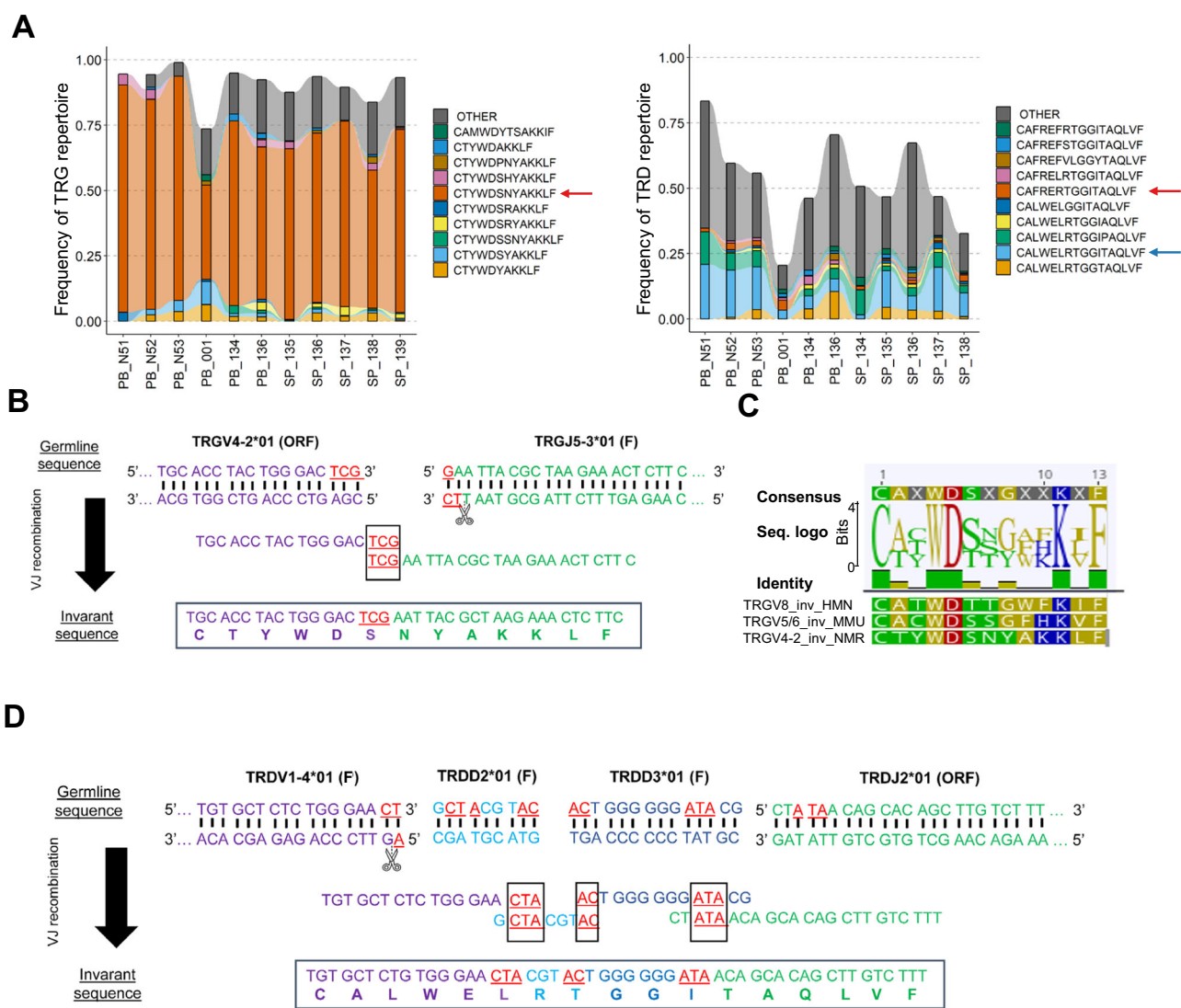

**Fig. 5 | Invariant naked mole-rat TRGV4-2 and TRDV1-4 CDR3 sequences can be generated by a mechanism of short homology repeat recombination. A** Alluvial plots displaying the frequency of top 10 shared gamma (left) and delta (right) TCR sequences in naked mole-rat animals. The red arrow in the left plot indicates the most abundant public TRG sequence. Blue and red arrows in the right plot indicate the most abundant public TRDV1-4 and TRDV1-6 sequences respectively. Sample identification numbers of individual naked mole-rats ($n = 11$) are indicated at the x axis; see Supplementary Table 1 for more information of each ID number. **B** Recombination using short homology repeats between TRGV4-2 and TRG5-3. **C** Sequence alignment and consensus logo plot (top) of the naked mole-rat TRGV4-2 invariant CDR3 sequence (NMR suffix), human TRGV8 invariant CDR3 sequence (HMN suffix) and the mouse TRGV5/TRGV6 invariant CDR3 sequence (MMU suffix). Amino acid key colors: dark yellow = non-polar aliphatic/aromatic, red = polar negative, blue = polar positive, green = polar neutral. **D** Recombination using short homology repeats between *TRDV1-4, TRDD2, TRDD3* and *TRDJ2*. **B, D** Overlapping nucleotides are shown in red. Source data are provided as a Source Data file.

CDR3 sequences were highly prevalent in cluster 5, which aligned with the T-cell γδ cluster reported in the original study based on gene expression profiles[41] (Fig. 6B, Supplementary Fig. 7A). Surprisingly, TRG and TRD CDR3 sequences were also enriched in cluster 4, previously[41] identified as a T-regulatory (T-reg)-like cluster, and a few sequences were present in cluster 3, corresponding to the original cluster[41] of CD8 T cells (Fig. 6B, Supplementary Fig. 7A, B). γδ T cells in cluster 5 exhibited differential expression of genes associated with cytotoxic lymphocytes (*TBX21, IL2RB, CX3CR1, CCL5, NKG7, SLAMF7, CCR5, FCGR3A*), and interestingly, some NK receptors (*KLRD1, KLRG1*), suggesting that cells from this cluster may possess the functional characteristics of bona fide NK cells in naked mole-rats (Fig. 6C, Supplementary Fig. 7C). Further confirmation of the NK-like phenotype of cluster 5 cells came from results of the hypergeometric enrichment test applied to the differentially expressed genes, as mouse NK-gene sets ranked among the top-associated terms in the analysis

(Supplementary Fig. 7D). Intriguingly, cells from cluster 4 displayed a transcriptome enriched with molecules associated with diverse functionalities, including members of the tumor necrosis factor receptor superfamily (*TNFRSF4, TNFRSF18*), as well as a set of genes associated with IL-17 production in mouse and human γδ T cells (*AHR, CD44, MAF, BLK*)[42–44] (Fig. 6C, Supplementary Data 1).

In sum, based on CDR3 as a natural barcode, we identified two functionally different γδ T cell clusters among blood naked mole-rat T cells: one associated with NK killing machinery while the other appears to be more functional diverse and include expression of type 3-associated genes.

**Naked mole-rat NK-like effector γδ T are enriched for invariant TCR sequences**

Analysis of the single-cell TCR data revealed that the two primary γδ T cell clusters (clusters 4 and 5) contained distinct TCR ecosystems: cells

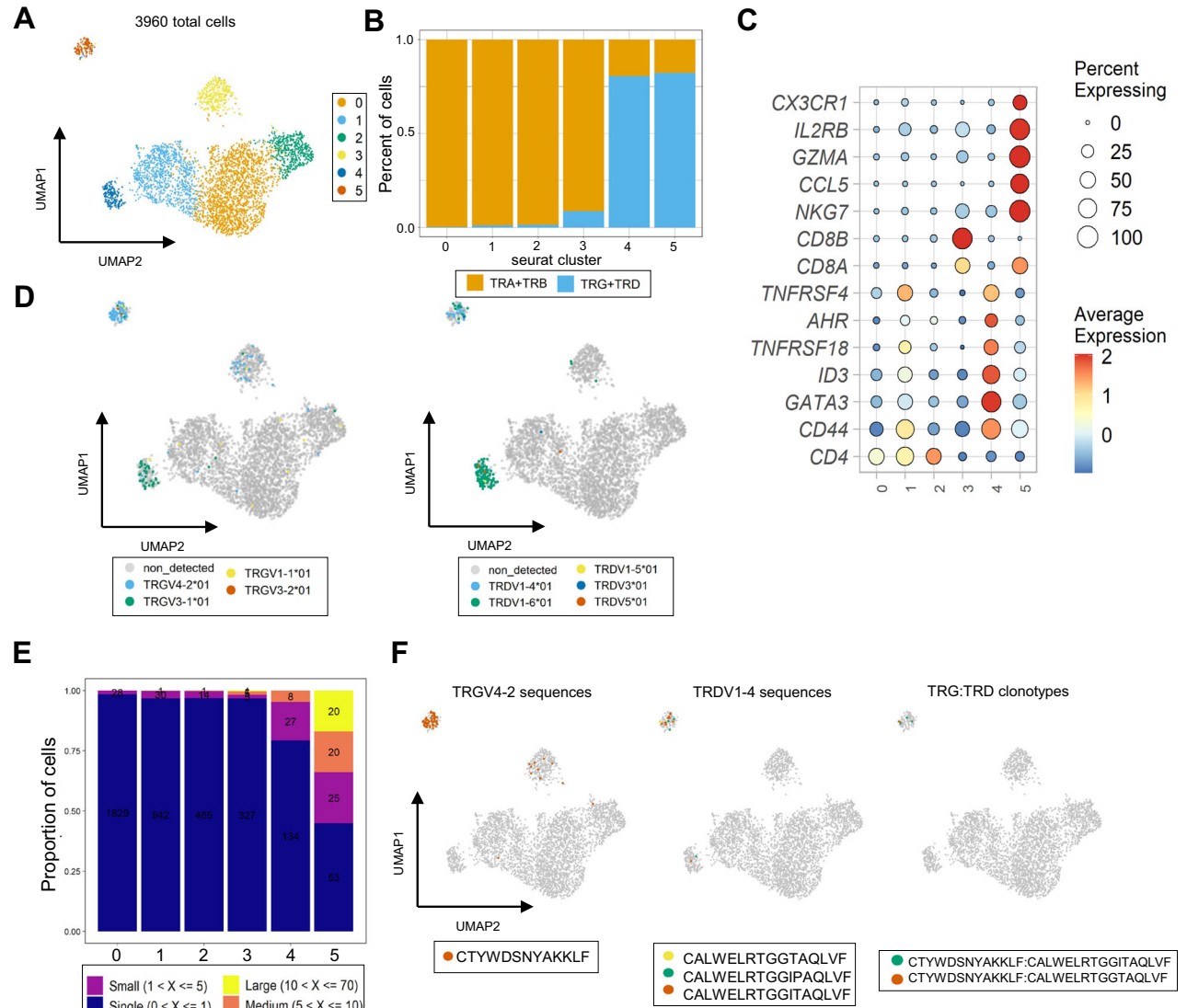

**Fig. 6 | Naked mole-rat NK-like effector γδ T are enriched for invariant TCR sequences. A** UMAP plot from blood T cells of 4 young and 4 middle-aged adult animals. The plot only shows single cells where TCR/CDR3 transcripts were detected. **B** Bar plots indicating the relative frequency of TCRαβ and TCRγδ cells in the different clusters identified in (**A**). Cells are classified in one of the two categories if at least one of both constitutive TCR chains (TRA or TRB and TRG or TRD) is detected in their transcriptome. **C** Dot plot heatmap displaying row-scaled Log2-fold change (logFC) expression values of selected DEGs in the different clusters identified in (**A**). Each dot represents the average expression profile of all the cells in the cluster. **D** UMAP plots (as generated under (**A**)) displaying TRGV (left) and TRDV (right) usage. **E** Clonality analysis of single cell TCR-seq data of the clusters identified in **A**. All TCR sequences (TRA,TRB,TRG and TRD) are classified in different categories according to their occurrences in the single cell dataset. Numbers in the bar plots refer to the number of TCR sequences in each category. **F** UMAP plots (as generated under (**A**)) highlighting specific TRGV4-2 (left) and TRDV1-4 sequences (middle) and TRG:TRD paired clonotypes (right). Source data are provided as a Source Data file.

from cluster 5 exhibited enrichment for TRGV4-2- and TRDV1-4-containing CDR3 sequences, while cluster 4 was predominantly composed of TRGV3-1- and TRDV1-6-containing CDR3 transcripts (Fig. 6D), in line with our bulk γδ TCR sequence data (Fig. 4B, Fig. 5). This variation in V gene usage was accompanied by differences in TRGJ and TRGC usage, with TRGJ5-3 and TRGC5 being prevalent in the TRGV4-2-enriched cluster 5, and TRGJ1-2 and TRGC1 in the TRGV3-1-enriched cluster (Supplementary Fig. 8A). Notably, these two TRGV-TRGJ-TRGC pairings are located in separate cassettes of the naked-mole rat TRG locus, although intriguingly, TRGV3-1-TRGJ1-2 are paired with the non-functional TRGC1 gene (Fig. 1A). All TRD sequences were paired with the same TRDJ gene (TRDJ2) and used the only available TRDC gene (Supplementary Fig. 8A, Fig. 1C). Differences in the level of TRG and TRD detection between clusters 4 and 5 (Fig. 6D) can be related to the fact the different types of γδ T cells can express different γδ TCR levels[45–48].

Clonality analysis of the TCR repertoires demonstrated that both cluster 4 and cluster 5 were enriched for expanded TCR sequences (Fig. 6E). Consistent with this observation, diversity was reduced in the TRG repertoire of both clusters 4 and 5 (Supplementary Fig. 8B). For TRD, however, this low level of diversity was limited to cluster 5 (Supplementary Fig. 8B). Visualization in the UMAP plot of cells expressing the invariant TRGV4-2 CTYWDSNYAKKLF sequence and the invariant TRDV1-4 CALWELRTGGTAQLVF sequence clearly demonstrated a localization bias towards the cluster 5 exhibiting NK-like effector features (Fig. 6F left and middle panel) and confirmed the pairing of the two invariant sequences (Fig. 6F right panel). Interestingly, we detected expression of the invariant TRGV4-2 sequence in the CD8 cluster (cluster 3); however, TRD detection was low, and the only detected sequences were non-invariant TRDV1-4-containing CDR3 sequences (Fig. 6D, F). Smaller expanded TRDV1-6 and TRGV3-

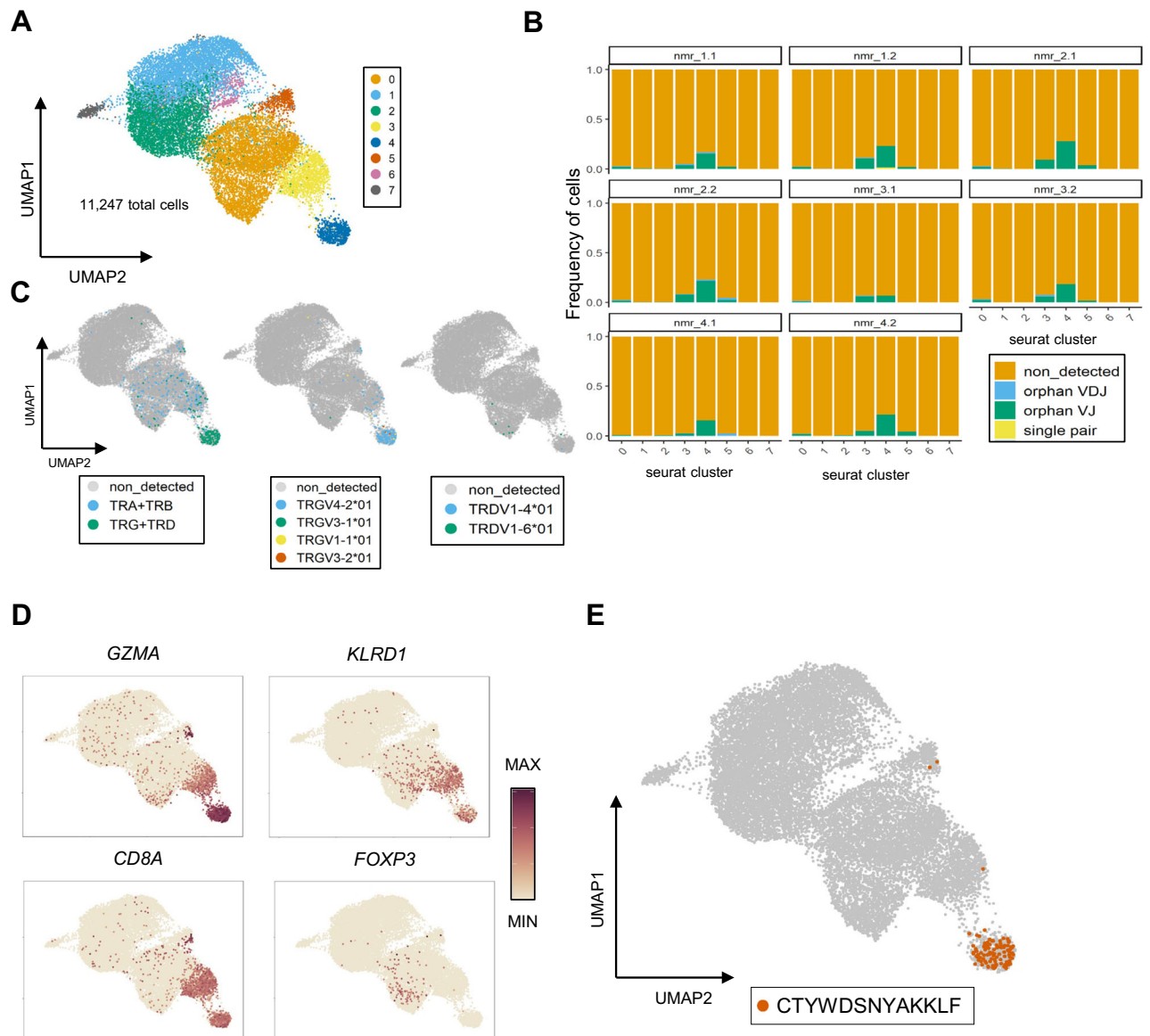

**Fig. 7 | Spleen NK-like effector γδ T cells also contain invariant TCRγ sequences. A** UMAP plot from spleen single cell T cells of 4 young-adult animals. **B** Bar plots indicating frequencies of detected TCR transcripts in the total T cell clusters identified in (**A**). Orphan_VDJ = cells with single TRB or TRD chain, orphan VJ: cells with single TRA or TRB chain, single pair: cells with a single TRA-TRB or TRG-TRD pair, non_detected: cells where TRUST4 did not extract any CDR3 information. Each animal has 2 replicate experiments labeled with the .1 and .2 suffix. **C** UMAP plots (as generated under (**A**)) displaying receptor subtype (left), TRGV (middle), and TRDV (right) usage (each graph shows the data of a single animal). **D** UMAP plots (as generated under (**A**)) displaying expression levels of selected genes. **E** UMAP plot (as generated under (**A**)) highlighting the cells that express the invariant TRGV4-2 sequence. Source data are provided as a Source Data file.

1 sequences were detected in the case of γδ T cells from cluster 4 compared with the invariant constituents of the NK-like TCR, with some of the TRDV1-6 sequences constituting private expansions (Supplementary Fig. 8C). Intriguingly, we detected two cells expressing the previously reported invariant TRDV1-6 sequence (Fig. 5A right panel, Supplementary Fig. 6A, B, Supplementary Fig. 8C) that clustered in cluster 5 (Supplementary Fig. 8C). Further analysis of the paired TRG-TRD data confirmed that cells from cluster 4 (enriched for *TRDV1-6*) preferentially paired with TRGV3-1 sequences in different animals, while cells in cluster 5 (enriched for *TRDV1-4*) exhibited pairing with TRGV4-2 sequences (Supplementary Fig. 8D). Lastly, the TRAV1-TRAJ33 MAIT-like sequence (Supplementary Fig. 6D) also clustered within the region where NK-like cells were present (Supplementary Fig. 8E).

In order to extend our findings in the blood to the spleen compartment, we re-analyzed a public available dataset from spleens of four young-adult animals (23–25 months old)[6]. After applying a similar pipeline as the one used before with the blood dataset, we observed lower levels of TCR detection in all T-cell clusters compared to the blood experiment (Fig. 7A, B). The majority of γδ T cells were localized in cluster number 4 and a few γδ T cells were scattered inside cluster 3 (Figs. 7A, C left panel). Cluster 4 corresponded to an NK-like effector cluster (Figs. 7A, D, Supplementary Data 2), and, similar to our findings in the blood dataset, the TRGV4-2 CTYWDSNYAKKLF CDR3 sequence was highly enriched in this cluster (Fig. 7C middle panel, Fig. 7E). Due to the extremely low detection levels, we were unable to draw conclusions regarding TRDV usage in this spleen cluster 4 (Fig. 7C right panel). Additionally, cells from cluster 3, which also contained some TRGV4-2 sequences, exhibited characteristics of conventional CD8 T cells (Fig. 7A, C, D), again aligning with our observations in the blood dataset (Fig. 6A, D, Supplementary Fig. 7A). Notably, the few TRDV1-6-containing sequences clustered far away from the TRGV4-2 cells,

within a region where the T-regulatory gene *FOXP3* is expressed (Fig. 7C, D).

Thus overall, the NK-like effector γδ T cells in the periphery (blood, spleen) express an invariant Vγ4-2/Vδ1-4 TCR, while in the blood another more variable, both at functional and TCR level, γδ T cell cluster is present.

### Naked mole-rat invariant γδ TCR sequences are generated in the thymus until mid-life

To gain insights into the natural dynamics of the γδ TCR repertoire across different age groups, we conducted TCR sequencing of thoracic and cervical thymic tissue samples[7] from neonates and young-adult (around 3 years old) animals. Despite the limited number of samples, we observed consistent trends in the general properties of the CDR3 regions as seen in the blood and spleen bulk TCR-seq results such as lower junctional diversity and shorter CDR3 sequences compared to the adult counterparts (Supplementary Fig. 9A). Analysis using Tree

maps revealed that in both age groups, the TRG and TRD repertoires exhibited oligoclonal patterns, whereas the TRA and TRB repertoires displayed high levels of polyclonality (Fig. 8A, B, Supplementary Fig. 9B). The invariant TRGV4-2 and TRDV1-4 sequences were detected in both age groups (Fig. 8A, B). This suggests that the production of these invariant sequences is sustained until 3 years of age, which contrasts with observations in mice and humans[9,43,49]. However, it cannot be excluded that the detected invariant sequences originated from the re-circulation of peripheral T cells towards the thymus[50]. To confirm the sustained production of these invariant TCR chains, we utilized a publicly available single-cell dataset obtained from the thoracic and cervical thymi of 2 young-adult (around 3 years old) and 2 middle-aged (around 11 years old) adult animals[7]. Cells from T cell clusters in the original thoracic and cervical thymic datasets integrated well with each other (Fig. 8C, D, Supplementary Data 3). The overall detection of TCR CDR3 transcripts using TRUST4 was at a similar level to what we observed in the blood single-cell dataset (Supplementary

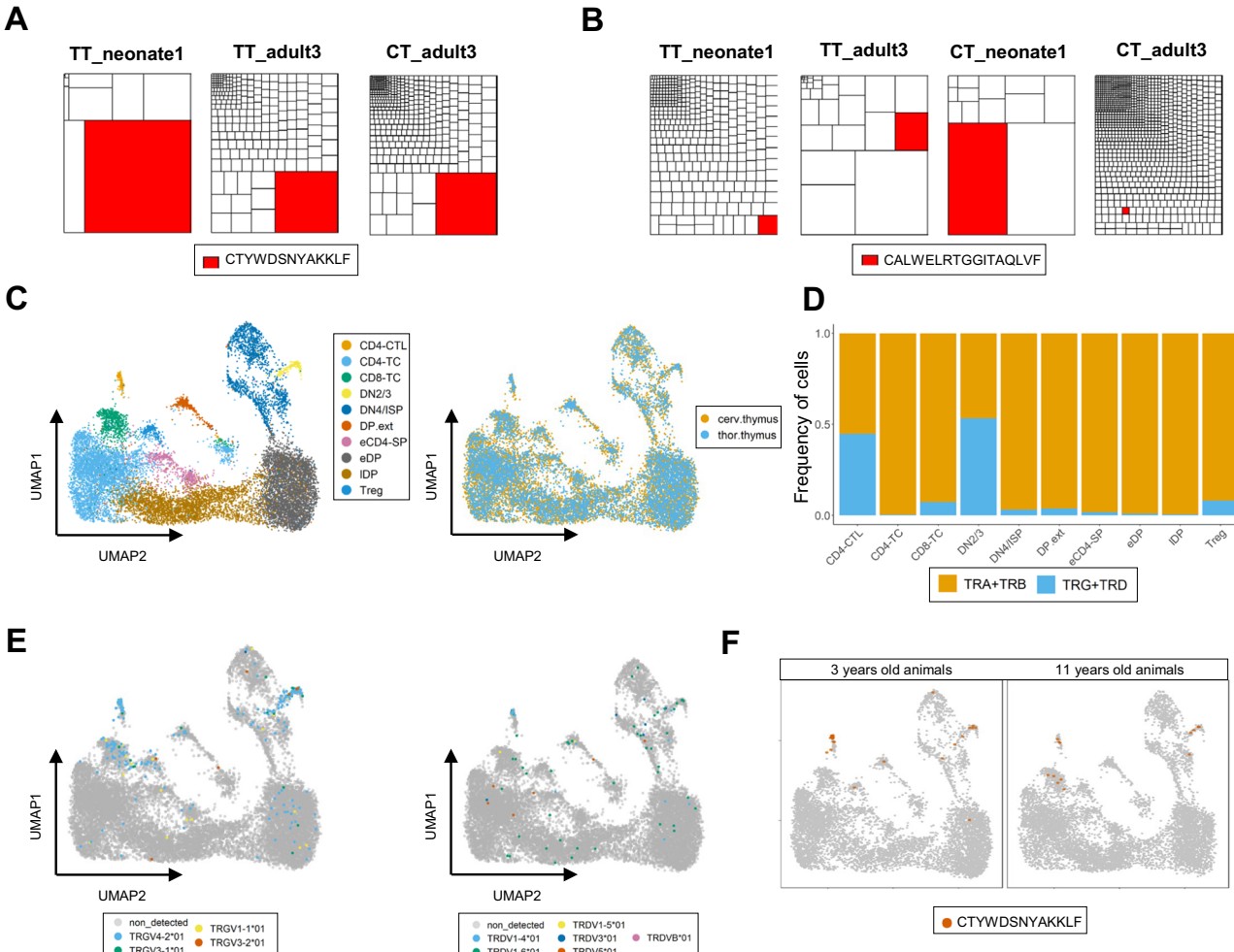

**Fig. 8 | Naked mole-rat invariant γδ TCR sequences are generated in the thymus until mid-life. A** Tree maps representing the TRG repertoire in neonate and adult thymic samples highlighting in red the prevalence of the invariant TRGV4-2 CDR3 sequence. **B** Tree maps representing the TRD repertoire in neonate and adult thymic samples highlighting in red the prevalence of the invariant TRDV1-4 CDR3 sequence. For (**A**, **B**): each Tree map represents the repertoire data of the indicated naked mole-rat ID (see Supplementary Table 1 for more information for each ID number). **C** UMAP plot colored based on identified clusters from thymic (thoracic and cervical) single cells (left) and colored by organ origin of the cells (right). Data are obtained from a public dataset with sequencing data from 4 animals (2 young-adult and 2 middle-aged adult individuals). **D** Bar plot indicating the relative frequency of TCRαβ and TCRγδ cells in the different thymic clusters identified under (**C**). Cells are

classified in one of the two categories if at least one of both constitutive TCR chains (TRA or TRB and TRG or TRD) is detected in their transcriptome. Cells without TCR transcript detection (non_detected) are not depicted in the plot (**E**) UMAP plots (as generated under (**C**)) displaying TRGV (left) and TRDV (right) usage in the single cell thymic data. **F** UMAP plot as generated under (**C**) highlighting the thymocytes that express the invariant TRGV4-2 sequence across the ontogeny. Abbreviations on (**C**): DN2/3 (double negative); DN4/ISP (immature single positive); eDP (early double-positive), lDP (late double-positive), eCD4-SP (early CD4 single-positive); CD4-TC (CD4⁺ T-lymphocyte); CD8-TC (CD8⁺ T-lymphocyte); DP.ext (extrathymic/circulating DPs); Treg (regulatory T cell); CD4-CTL (CD4⁺ cytotoxic T-lymphocyte). Source data are provided as a Source Data file.

Fig. 9C). However, the relative abundance of γδ T cells in T-reg and NK-like effector clusters (referred to as CD4-CTL in the original paper's annotation) was lower (Fig. 8D, E). Interestingly, a significant proportion of γδ TCR transcripts, primarily TRG chains, were detected in the double negative (DN)2/3 cluster, which corresponds to T-cell precursors (Fig. 8D, E). In contrast, the detection of TRA rearrangements in this cluster was relatively low compared to TRG, TRB, and TRD transcripts, which follows the stereotypical sequential loci rearrangement described in human and mouse thymocyte development (Supplementary Fig. 10A)[13]. Consistent with the findings in blood and spleen, TRGV4-2 transcripts were particularly abundant in the NK-like effector cluster (annotated as "CD4-CTL", Figs. 8C, E left panel). Due to lower detection of TRD transcripts, the presence of TRDV1-4 sequences in that cluster was limited (Fig. 8E right panel). As observed in other anatomical contexts, invariant TRGV4-2 sequences were highly enriched in the NK-like effector thymic cluster at 3 and 11 years and, interestingly, they were already present in the progenitor DN2/3 cluster at both 3 and 11 years of age (Fig. 8F), indicating sustained production of the invariant CTYWDSNYAKKLF sequence until middle-aged adulthood. Similarly, while the levels of detection of the invariant TRDV1-4 sequence were quite low, they were only detected in the CD4-CTL cluster (Supplementary Fig. 10B left panel). Furthermore, akin to T cells in blood, sequences of the TRAV1-TRAJ33 MAIT-like sequence were present in the NK-like cluster, with some sequences also detected in the maturing cluster eCD4-TC (Supplementary Fig. 10B right panel). Expression of naked mole-rat *MR1*, the MHC-like molecule involved in the development and function of MAIT cells in mice and human[40], could be observed (Supplementary Fig. 10C) and it showed high homology with its mouse and human counterparts (Supplementary Fig. 10D, E).

## Discussion

The naked mole-rat lack NK cells[6]. Here, we identify public invariant γδ T cells that are pre-programmed during development in the thoracic and cervical thymi of the naked mole-rat with NK-like effector functions, before going to the periphery (blood, spleen).

The invariant TRG chain is formed by the rearrangement in the genome (V(D)J recombination) of the *TRGV4-2* gene with the *TRGJ5-3* gene while the invariant TRD chain is formed by the rearrangement of TRDV1-4/TRDD2/TRDD3/TRDJ2. We identified in all these sets of genes short homology repeats at the coding ends, also known as microhomology domains, that likely drive the formation of the invariant TRG and TRD chains. These invariant sequences dominate the naked mole-rat γδ TCR repertoire and by single-cell analysis we formally show that these chains pair and thus assemble a complete invariant Vγ4-2/Vδ1-4 γδ TCR. Such short homology repeats are used by early life γδ T cells both in mouse and human[8]. In these species the generation of these TCRs is limited to the fetal period, when the expression of TdT is low or absent, thus allowing the recombination of the short homology repeats to occur as there is not disturbance by the random addition of N nucleotides at the coding ends of the genes[8,51]. Remarkably, in contrast to mouse and human[9,43,49], in the naked mole-rat this invariant Vγ4-2/Vδ1-4 TCR is being generated at least till adult life (11 years old). This might be due to the relative low level of N additions that are seen at that age: for example for the TRG chain this is around 0.5 for blood/spleen and 2 for thymus in the naked mole-rat, while for human adult blood/post-natal thymus (including a 9 year-old thymus) the number of N additions is 5, thus up to 10 times higher[49,52].

We propose that the naked mole-rat invariant Vγ4-2/Vδ1-4 TCR recognize a common molecular signal shared by naked mole-rat cells undergoing transformation which then activates the NK-like killing effector program (including release of perforin and granzymes) that is intrinsically present in the peripheral Vγ4-2/Vδ1-4 γδ T cells prior to signal encounter. Such functional programming is also observed in the mouse and human fetal thymic invariant γδ T cells[14,43,49], but only in the

naked mole-rat, this NK effector program persists following the production of the Vγ4-2/Vδ1-4 thymocytes at least till 11 years. This allows the continuous generation of "fresh" invariant γδTCR NK-like effector T cells that can actively control the formation of cancer cells in the periphery. Such continuous production would prevent exhaustion of γδ T cells as has been seen for example when attempting to activate multiple times human Vγ9Vδ2 T cells in clinical trials[27]. Because of this, efforts are made to generate γδ T cell cellular products for adoptive cellular immunotherapy starting from human cord blood that results in larger expansions compared to adult peripheral blood because of their less differentiated nature[53]. In human, the prevalent TRDV gene used in early life is *TRDV2*, which can pair with *TRGV9* to form innate phosphoantigen-reactive Vγ9Vδ2 T cells or pair with other Vγ chains in order to generate Vγ9-Vδ2+ γδ T cells[8]. Human Vγ9Vδ2 γδ T cells can also be made in the post-natal thymus, but with a different type of Vγ9Vδ2 TCR and at much lower frequencies[54,55]. Remarkably, while in the mouse no orthologues for the human *TRGV9* and *TRDV2* genes could be found, we showed here, using unrooted phylogenetic analysis, that clear orthologue genes exist in the naked mole-rat. However, these were categorized as pseudogenes, and we could indeed not find any usage of these TRGV and TRDV genes within expressed CDR3 sequences. In parallel, no members of the BTN3 group of butyrophilins, important for phosphoantigen-reactivity of the human and alpaca Vγ9Vδ2 TCR[12,32,33,56], could be identified in the naked mole-rat, which is in line with the proposed co-evolution of functional TRGV9, TRDV2, and BTN3 genes[12,57]. It is therefore unlikely that the naked mole-rat possess phosphoantigen-reactive γδ T cells. We propose that the role of innate γδ T cells is rather taken by the abundant subset of invariant Vγ4-2/Vδ1-4 γδ T cells. The naked mole-rat *TRGV4-2* gene was most similar to mouse *Trgv3* (annotated as an open-reading frame, IMGT) and *Trgv1/Trgv2* that are used in the adaptive mouse γδ TCR repertoire[14]. There were also some similarities with the human *TRGV11* gene, but this gene is a pseudogene[58]. Thus, there appears to be rapid evolution of the TRGV genes, which is also observed for the functional TRGV genes (with the noted exception of *TRGV9*) within primates[59] and other mammals[60]. Despite the absence of obvious similarities between naked mole-rat, mouse, and human regarding the TRGV genes that are used in the formation of invariant γδ TCRs, it was interesting to observe that at the CDR3 level there was a high homology, both at the level of CDR3 length and amino acid physicochemical properties, of the naked mole-rat TRGV4-2 CDR3 sequence CTYWDS-NYAKKLF with the human fetal-derived TRGV8 CDR3 CATWDTTGWFKIF[36,43,49] and mouse fetal-derived Trgv5/v6 CDR3 CACWDSSGFHKVF[37,38,52]. This may indicate that these invariant CDR3γ sequences may be involved in the recognition of similar molecular moieties, that are for example expressed on malignant cells. The strong bias of the naked mole-rat TCR repertoire towards an invariant clonotype may also reflect the lower need for a high polyclonal TCR repertoire due to a possible lower pathogen diversity in the underground environment. It has been suggested that naked mole-rat's unique living conditions have led to the acquisition of intrinsic anti-cancer mechanisms as a secondary effect. Similarly, the pathogen-deprived environment in which they reside may have facilitated the emergence of invariant TCR mechanisms for protection, resembling the human fetal context where the pathogen-privileged utero environment also relies on an invariant γδ TCR repertoire to combat congenital infections that breach its protected sterile environment[36,61,62].

Besides the invariant Vγ4-2/Vδ1-4 NK-like effector cells, we identified in the blood of the naked mole-rat also a γδ cell cluster that showed a more variable TCR and functional repertoire. While there was enrichment for the usage of TRGV3-1 and TRDV1-6, the CDR3 regions were more variable than in the Vγ4-2/Vδ1-4 subset, especially for the TRDV1-6 CDR3. TRGV3-1 paired with a different TRGJ (TRGJ1-2) than the "invariant" TRGV4-2 and, intriguingly, with the *TRGC1* gene which was annotated as a constant pseudogene because of the lack of exon 2.

However, since no other V gene could be observed in the TRGV3-1/TRGJ1-2/TRGC1-expressing cells, we conclude that this TRG chain is likely the functional chain that pairs with the TRDV1-6 chain. Of note, a similar usage of a TRGC lacking exon 2 in an expressed TRG chain has been described in salmon[63]. More generally, different TRGC can show different exon-intron organizations, which is mainly linked to the different number of exon 2 copies, of which the biological significance remains unclear[60]. Functionally, the Vγ3-1/Vδ1-6 γδ T cells were clearly different from the NK-like program observed in the invariant Vγ4-2/V1-4 cells. They express a range of genes without clear association to a particular programmed function, although some genes linked to IL-17 production were enriched.

We could not detect TCR sequences that are similar to the ones described to be present in invariant NKT cells in other species, which coincided with the absence of *CD1D* from the naked mole-rat genome, the antigen-presenting element that is essential for the development and activation of these unconventional αβ T cells[40]. In contrast, a low level (0-0.4% of TRA sequences) of a MAIT TRA sequence was detected, together with *MR1*, the presenting element of vitamin B metabolites to the MAIT TCR[40]. Thus, while the blind mole-rat (*Spalax galili*) lacks both iNKT and MAIT cells[64], the naked mole-rat conserved low levels of MAIT cells. The absence of iNKT and only low levels of MAIT cells in the naked mole-rat could explain the high prevalence of γδ T cells, since in the mouse these three unconventional T cell types compete for the same niche[65] and we showed that the few MAIT cells detected in our single-cell data cluster functionally together with the invariant Vγ4-2/Vδ1-4 γδ T cell cluster suggesting the γδ and MAIT cells also share a functional niche in the naked mole-rat.

In summary, we have identified and characterized γδ T cell biology in the naked mole-rat providing a framework to investigate the intrinsic anticancer biology of this species from an immunological γδ T cell-based perspective.

## Methods

### TCR locus annotation

Two chromosome-level *Heterocephalus glaber* genome assemblies, the naked mole-rat maternal (GCA_944319715.1) and paternal (GCA_944319725.1), were obtained from the National Library of Medicine (NCBI) Assembly database to study the TR loci[66]. Maternal and paternal assemblies were derived from the gonad tissue of a 5-year-old male naked mole-rat generated via the 10x Linked reads sequencing and Nanopore long-read sequencing respectively. The annotation of the T-Cell Receptor (TR) loci of the *Heterocephalus glaber* was conducted following the comprehensive IMGT® biocuration pipeline[67], which is guided by the axioms and concepts of IMGT-ONTOLOGY. These concepts include "IDENTIFICATION," "DESCRIPTION," "CLASSIFICATION," "NUMEROTATION," "LOCALIZATION," "ORIENTATION," and "OBTENTION"; in order to maintain the accuracy and consistency of the annotation process by providing a controlled vocabulary and annotation rules[68].

To identify and extract the TR loci from the corresponding chromosomal assembly, a comparison was performed between them and the TR reference sequences of *Homo sapiens* using the Basic Local Alignment Search Tool (BLAST)[69]. The delimitation for each locus was determined by detecting the IMGT® Bornes, conserved non-TR-coding genes located upstream of the first or downstream of the last gene of an IG or TR locus[28]. The IMGT® 5' bornes for the TRB, TRG, and TRAD locus are *MOXD2*, *AMPH*, and *OR10G3* respectively, while their IMGT® 3' bornes are *EPHB6*, *STARD3NL*, and *DAD1* respectively. Each IMGT® annotated locus sequence was submitted to the Third Party Annotation (TPA) database and was assigned a unique NCBI accession number. In the naked mole-rat maternal assembly, the accession numbers are as follows: BK064754.1 for TRA/TRD, BK064759.1 for TRB, and BK064756.1 for TRG. In the naked mole-rat paternal assembly, the accession numbers are: BK064758.1 for TRB and BK064755.1 for TRG.

Note that we did not submit a naked mole-rat annotation for the TRAD locus based on the paternal assembly because of the presence of gaps, indicative of lower quality sequencing in that genome region.

The IMGT® nomenclature for the TR genes, was based on the "CLASSIFICATION" concept of the IMGT-ONTOLOGY[68]. TR genes of the naked mole-rat were classified into a subgroup based on sequence identity with human and/or mouse TR subgroups, calculated using IMGT V-QUEST[70], Clustal Omega[71], and Nphylogeny[72]. Variable genes with a V-REGION identity score above 70%, while those with scores below 70% were assigned to new subgroups. TR variable genes were marked by the subgroup number, and if there were multiple genes belonging to the same subgroup, they were followed by a hyphen and a number indicating their relative position in the locus. Genes with highly degenerated V-REGION that cannot be classified into a subgroup were marked with a letter instead of a subgroup number. Additionally, new alleles were annotated when the coding part of each gene (V-REGION, J-REGION, D-REGION or C exons) had an identity score of ~99% in different assemblies. The functionality of genes and alleles was defined according to the "IDENTIFICATION" axiom of IMGT-ONTOLOGY. An allele was considered as functional (F) if its coding region had an open reading frame without stop codons or any defects in the splicing sites, recombination; as open reading frame (ORF) if its coding region had an open reading frame without stop codons but showed alterations in the splicing sites, recombination signals or conserved amino acids; and as pseudogene (P) if the coding region had stop codon(s) and/or frameshift mutation(s)[73].

The annotated data of the current analysis were entered into the IMGT® reference directory and IMGT® databases, including IMGT/LIGM-DB[74], IMGT/GENE-DB[75], IMGT/3Dstructure-DB[76] and IMGT/2Dstructure-DB[28] by using a set of standardized labels described in the "DESCRIPTION" axiom of IMGT-ONTOLOGY. The data also entered in IMGT® sequence analysis tools (IMGT/V-QUEST (6), IMGT/HighV-QUEST[77], and IMGT/DomainGapAlign[78]. Finally, all the genomic annotated information of the TR loci of the naked mole-rat was consolidated into the IMGT® Repertoire (http://www.imgt.org/IMGTrepertoire/), which encompasses various web resources such as: Locus representation, Locus description, Locus in genome assembly, Locus gene order, Locus Borne, Gene tables, Potential germline repertoire, Protein displays, Alignments of alleles, Colliers de Perles, and germline [CDR1-IMGT.CDR2-IMGT.CDR3-IMGT] lengths. The proposed nomenclature is considered provisional based on future data being available and involvement of IUIS at the appropriate time/level.

### Phylogenetic tree construction and gene alignment

Maximum Likelihood trees for each TCR locus and BTN(L)/SKINT proteins were reconstructed with the human, mouse, and naked mole-rat datasets, using the MEGA (Molecular Evolutionary Genetics Analysis) software tool (version 11.0.13)[79]. More specifically, the TCR loci phylogenetic trees were based on nucleotide multiple sequence alignments (MSA), generated with representatives of V-REGION sequences from each subgroup, with the exception of the highly degenerate pseudogenes (TRAVA, TRBVA, TRAVB, etc). The MSA was performed with the online multiple sequence alignment program MAFFT (version 7)[80], with default parameters, featuring automatic algorithm selection, a 200PAM/K = 2 scoring matrix, a gap opening penalty of −1.53, and a gap extension penalty of 0.0, to achieve high alignment accuracy and consistency. The tree visualization was performed using iTOL v6[81], and the different human subgroups were represented with distinct colors.

The BTN(L)/SKINT trees were reconstructed using MAFFT (version 7) with amino acid multiple sequence alignments of the BTN(L) and SKINT proteins extracted from NCBI. Default parameters were applied in MAFFT, using automatic algorithm selection, the BLOSUM62 scoring matrix, a gap opening penalty of -1.53, and a gap extension penalty of 0.0. A single representative sequence for each

protein family was selected for inclusion in the analysis. The tree was visualized with iTOL v6 and the colors were assigned manually indicating the orthologous proteins identified using the NCBI orthology database.

For the alignment and comparison of *MR1* genes, the Clustal Omega multiple sequence alignment package (version 1.2.4)[71] was used, with default parameters, utilizing the Gonnet transition matrix, a gap opening penalty of 6 bits and a gap extension penalty of 1 bit for the gene sequence alignment. The results visualization was achieved with Jalview (version 2.11.3.2)[82]. In the alignment of invariant CDR3 γ regions from human, mouse, and naked mole-rat Geneious Prime (version 2023.1.2) was used to group sequence by similarity using Clustal Omega method (version 1.2.4).

## Animals and sample collection

Tissue samples (blood, spleen, and cervical/thoracic thymi) were collected from animals at different developmental stages: neonates 1st day of birth, young-adult animals (around 3 years old), and middle-aged adults (around 11 years old). The primary factor motivating the choice of spleen and blood for investigation of the TCR repertoire in peripheral tissues was the feasibility of obtaining these tissues in the naked mole-rat at various developmental timepoints, including neonates. The cervical and thoracic thymi were included in the study as they are the T cell-generating organs in the naked mole-rat. All neonates in the study were littermates, while young adults were sourced from three different litters: adults 1, 2, and 3 were littermates, adults 5 and 6 were also littermates, and adult 4 was unrelated to the others (IDs correspond to Supplementary Table 1). The only middle-aged animal (Adult 7) was from an independent litter. After collection tissue samples were lysed in RLT + buffer and stored at −80 °C for further analyses. Ethical and legal approval was obtained by the University of Rochester Committee on Animal Resources (UCAR), and animal experiments were performed in accordance with guidelines instructed by UCAR with protocol numbers 2009-054 (naked mole rat) and 2017-033 (mouse). Naked mole rats were housed at the University of Rochester colonies, breeding and detail described elsewhere[83].

## Bulk TCR-sequencing and analysis

RNA was isolated from tissues single-cell suspensions with the RNeasy Micro Kit (Qiagen). cDNA was generated performing a template switch anchored RT-PCR. RNA was reverse transcribed via a template switch cDNA reaction using in the same tube **1)** a common TRGC primer with degenerate nucleotides specific for all the possible functional TRGC genes identified with IMGT annotation rules (5′-GAAAGRTATGTYC-CAGCCTY-3′), **2)** a TRDC primer (5′-GAGACAAGCAACCTTTGTTCCA TT-3′) to amplify CDR3δ sequences, **3)** a template-switch adapter (5′-AAGCAGTGGTATCAACGCAGAGTACATrGrGrG-3′) and **4)** the Super-Script II RT enzyme (Invitrogen). For CDR3α and CDR3β sequences, we used oligodT primer (5′-TTTTTTTTTTTTTTTTTTTTTTTTTTTTTTVN-3′) instead of specific locus primers to generate cDNA. The cDNA was then purified using AMPure XP beads (Agencourt). Subsequent amplification of the TRG and TRD region was achieved using a combination of TRGC specific primers including 1) a TRGC2 primer (5′-*GTCTCGTGGG CTCGGAGATGTGTATAAGAGACA*GAAAGGGCTTGGGGGAAATGTT-3′), 2) a TRGC3 primer (5′-*GTCTCGTGGGCTCGGAGATGTGTATAAGAGACA*G GCTTGGGGAAAATGTCTGTGTC-3′) and 3) a TRGC5 primer (5′-*GTCTC GTGGGCTCGGAGATGTGTATAAGAGACA*GTGGGGGAAATGTCTCCAT CTG-3′). These combination of TRG primers was mixed as well with a specific TRDC primer (5′-*GTCTCGTGGGCTCGGAGATGTGTATAAGA-GACA*GTGGTTTGGAAGGAGGCTGAAT-3′, adapter in italic) and a primer complementary to the template-switch adapter (5′-*TCGTCGGC AGCGTCAGATGTGTATAAGAGACA*GAAGCAGTGGTATCAACGCAG-3′, adapter in italic) with the KAPA Real-Time Library Amplification Kit (Kapa Biosystems). In the case of TRA and TRB sequence the KAPA enzyme and the primer complementary to the template-switch

adapter were combined with a TRAC specific primer (5′-*GTCTC GTGGGCTCGGAGATGTGTATAAGAGACA*GCTTTCAGCTGGTACACGG CA-3′) and a TRBC specific primer (5′-*GTCTCGTGGGCTCGGAGATG TGTATAAGAGACA*GATGGCTCAAACACAGCCACC-3′). After purification with AMPure XP beads, an index PCR with Illumina sequencing adapters was performed using the Nextera XT Index Kit. This second PCR product was again purified with AMPure XP beads. HTS of the generated amplicon products was performed on an Illumina MiSeq platform using the V2 300 kit, with 300 bp at the 39 end (read 2) and 150 bp at the 59 end (read 1) (at the GIGA center, University of Liège, Liège, Belgium). Default parameters were used except to assemble TRDD gene where three instead of five consecutive nucleotides were applied as assemble parameter. Raw sequencing reads from fastq files (read 1 and read 2) were aligned to the constructed IMGT reference for V, D, and J genes from naked mole-rat (based on the long-read chromosome assemblies as described under "TCR locus annotation") using MiXCR software version 4.1.2[84]. Default parameters were used except to assemble TRDD gene where three instead of five consecutive nucleotides were applied as assemble parameter. CDR3 sequences were then exported and analyzed using VDJtools software version 1.2.1 using default settings[85]. Sequences out of frame and containing stop codons were excluded from the analysis. The degree of TCR repertoire overlap between two different samples was analyzed using the overlap F metrics. The *F* value is the geometric mean of relative overlap frequencies and is thus calculated as follows: $F_{ij} = \sqrt{f_{ij}f_{ji}}$ ; where $f_{ij} = \sum_{k=1}^{N} \varnothing_{ik}$ is the total frequency of clonotypes that overlap between samples *i* and *j* in sample *i*. Tree maps were created using the Treemap Package (version 2.4–3). Visualization of gene usage, public clonotype frequencies, and clonality were generated using Immunarch R package (version 1.0.0)[86]. Hierarchical clustering based on V gene usage was done based on Jensen–Shannon divergence. Visualization of gene usage, public clonotype frequencies and clonality were generated using Immunarch R package (version 0.9.0)[86]. Hierarchical clustering based on V gene usage was done based on Jensen–Shannon divergence.

## TCR extraction from public single-cell RNA-seq datasets

In order to construct single cell TCR-seq libraries from public naked mole-rat single cell RNA-seq data[6,7,41] sequencing fastq files were processed using TRUST4 software (version 1.0.10)[87]. Naked mole-rat TCR reference from IMGT was used as -f and --ref parameters when using "run-trust4" function. TRUST4 output in AIRR format was used as input for Scirpy package (version 0.13.0)[88] to re-construct single cell paired TCR repertoires based on barcode sequences.

## Single-cell RNA-seq data processing

Publicly available processed Seurat objects (with qc analysis already performed) from single cell RNA-seq data were downloaded from the corresponding public repositories[6,7,41] (available in the GEO database under accession codes GSE185721, GSE202903 and GSE132642). The animals used in these public single cell RNA-seq repositories are different animals, derived from different litters, than the ones used for the bulk TCR experiments described in the current study. All downstream single cell analyses were performed using the R package *Seurat* version 4.3.0[89]. Single cell TCR libraries were added as metadata to the Seurat objects. In all 3 datasets T-cell clusters were subset based on the annotation obtained from the original studies and subsequent analysis were performed on them. In the case of blood data due to the high TCR-extraction efficiency of the TRUST4 pipeline, T-cell clusters were further subset based on the presence of TCR transcripts containing a rearranged CDR3 sequence. In all cases after subset of data principal components (PCs) were calculated using "RunPCA" and 30 dimensions were chosen as input for "RunUMAP" function. UMAP representation was used to generate bidimensional coordinates for each cell. The

k-nearest neighbors of each cell were determined using the ´Find-Neighbors´ function and this knn graph was used to construct the shared nearest neighbor (SNN) graph by calculating the neighborhood overlap (Jaccard index) between every cell and its k.param nearest neighbors. Finally, the ´FindClusters´ function was used to cluster cells using the Louvain algorithm based on the same PCs as RunUMAP function. Differential gene expression analysis comparing gene expression of each cluster to all the others was performed by the "FindAll-Markers" function using Wilcoxon-Rank sum test method (two-sided). DEGs were selected based on a log2-fold change (logFC) $\geq 0.2$ difference between the average gene expression of the clusters, a percentage of expression equal to or greater than 15% in at least one test cluster (min.pct $\geq 0.15$), a difference equal to or greater than 15% in the fraction of detection between the two groups (min.diff.pct $\geq 0.15$) and adjusted $p < 0.05$ (based on Bonferroni correction using all genes in the dataset). Diversity and clonality analysis of single cell TCR-seq data were done thanks to *scRepertoire* package (version 1.11.0)[90].

### Hypergeometric test analysis

Differential expressed genes from blood NK-like effector cluster were used as input for enrichr function of *enrichR* package (version 3.0.1)[91]. Tabula Muris gene sets were used as reference to perform the comparative test.

### Statistical analysis

Statistical analyses were performed using GraphPad Prism 8.0.2 software or with Immunarch package depending on the nature of the data. Two-tailed Student's *t* test were used for normally distributed data; Mann-Whitney U and Wilcoxon matched paired tests were used for nonparametric data. Differences between more than two groups were analyzed using Kruskal−Wallis and Dunn's post-tests for nonparametric data.

## Data availability

The raw fastq sequencing data of TRA, TRB, TRG, and TRD bulk TCR repertoires data generated in this study have been deposited in the in the SRA public database under accession code PRJNA998815. Source data are provided with this paper.

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

## Acknowledgements

This work was supported by an "Actions Blanches" grant (ULB; DV), a WELBIO investigator program (WELBIO-CR-2022 A– 15, D.V.) and by US National Institutes of Health grants to A.S. and V.G. G.S.S. is supported by Télévie/FNRS, WELBIO and "Actions Blanches". We thank Isabel Kuppens to bring the naked mole-rat to our attention. We thank Alexandre Albani for contribution to the biocuration of the naked mole-rat loci, Veronique Giudicelli, Nika Abdollahi for repertoire analysis and discussions, Geraldine Folch and Joumana Michaloud for data validations as well as the whole IMGT team.

## Author contributions

G.S.S. designed and performed experiments, analyzed data, and wrote the manuscript; S.E. designed and performed experiments; M.G., A.P., and S.K performed analysis of genomic data; V.G. and A.S. provided samples and resources; D.V. designed and supervised the research, provided resources, analyzed data and wrote the manuscript; all co-authors read and approved the manuscript.

## Competing interests

The authors declare no competing interests.
