## [Peer Review File · Nature Communications]

Invariant $\gamma\delta$ TCR natural killer-like effector T cells in the naked mole-ratREVIEWER COMMENTS

Reviewer #1 (expert in annotation and assembly of antigen receptor loci):

In this manuscript, Sanchez et al. have conducted an analysis of the T cell receptor (TCR) repertoire in the naked mole-rat, including: (1) germline gene annotations for each TCR locus in this species, (2) analysis of alpha, beta, delta and gamma TCR transcripts, and (3) focused analysis of single-cell RNA transcriptomes associated with invariant gamma/delta natural killer-like effector T cells. The motivation for this analysis is driven by a need to better understand T cell mediated immunity in this unique species. Overall, the manuscript is well-written and easy to follow. Detailed concerns/comments are provided below.

Line 62: should be "gene".

Line 67: "at 'the' protein level"

Line 92: It is not exactly clear that this study "shows" that these cells have cytotoxic effector functions, but maybe more specifically that these cells have a specific marker/transcriptomic phenotype. The wording should be tempered here.

Line 98: The legend for Figure 1 does not sufficiently explain features of the figure. Much more detail is needed. In the figures/maps themselves, it is not clear what scaffolds are being used for which parts of the maps, yet the authors say that more than one scaffold is used. And the methods claims that two different assemblies are used even, but there is no indication here what assemblies are represented in these maps. What do the double slanted lines represent. Are these "gaps" in the assembly/scaffolds? What are the black arrows under assembly in panel B? Why are the genes above those arrows named differently? Are they TRA genes?

Line 110: Instead of the use of "borne", the authors should consider using terms common to the genomics field to explain this. For one, it hasn't yet been defined by this line in the manuscript, but it seems the authors mean the nearest 5' non-TR flanking gene in the assembly/scaffold. To improve readability/comprehension, the authors should probably just say that instead of using uncommon jargon.

Line 110: Check Nat Comm conventions whether specific gene names should be italicized.

Line 127: The authors mention that some genes have "provisional" names. It's not clear what is mean by this, and based/on compared to what. In addition, this is confusing as the maps shown in Figure 1 look to be claiming that all names are "provisional". Not clear what this means, how it was decided upon, or how to interpret.

Line 131: Not clear what 'invariant' means in this section. Can you define what is meant by invariant when referring to germline genes? And why it is relevant to the phylogenetic analysis in this section? Is this a term for germline genes that contribute to 'invariant' TCRs? If so this should be stated, and also important to clarify whether there are TRG/D genes that are not 'invariant'.

Lines 136-138: Why are those genes exceptions? And how were exceptions decided upon?

Line 147: How is "high homology" defined?

Figure 2/S1: Legend needs citations for MEGA, MAFFT, etc.

Lines 150-151: Can the authors provide support for this claim beyond just the assembly data?

Were these genes also not observed in the expressed TCR repertoire?

Line 155: This claim should be tempered. Fair to say, no clear orthologue was observed in the assemblies you've analyzed, but not extrapolating to the species as a whole might be a leap.

Lines 156-162: This seems out of place given the header of the results section. Its relevance should be better motivated, perhaps written into its own section.

Line 164: At the beginning of this section, the authors should state which genes from the annotations in Figure 1 are supported at nucleotide level from TCR transcript reads from the animals analyzed.

Line 172 (Figure 3B): Why did the authors combine adult blood and spleen for this analysis?

Line 173: Believe it should be "tendency".

Lines 166-168: The number of animals studied, and their origins should be stated here in text (rather than make reader go methods or figure legends). Are they unrelated animals, littermates, etc.?

Figure 3B: What is the basis for including the genes that are plotted. Usage levels? Is the reader to assume that all other genes have no usage? It would also be useful to comment on the fact that 3 of the 4 genes shown are annotated as ORF, rather than functional. What is the basis for that?

Line 167: What is the rationale for studying these tissues?

Line 177-179: The authors should dissect this claim. What is it based on? If the authors have simply run a statistical test in each group separately and then made this assessment based on individual P values, may not be appropriate. Authors should be more explicit.

Line 190: Metric used for "sharing" (F value) should be stated here in results. "Repertoire sharing" is vague.

Figure 4 legend: typo? "TRG5-3"

Figure 5D: Could the authors comment on low detection of TRGV3-1 in cluster 4? It seems that the frequency of detected gamma TCRs is low relative to delta? The inverse seems to be somewhat true for clusters 3 and 5. How have the authors verified they haven't artificially missed some unidentified TCR chains?

Line 297: number of animals should be stated.

Line 299: Figure 6B, no x axis labels. Looks as though it should be clusters from panel A.

Lines 325: number of animals included in analysis should be stated.

Figure S6C: Potential typo. Should it be "other" rather than "ather"?

Line 427: Typo. "TRCG"

Line 459: The authors should better outline the source of the scaffold assemblies used so it is clear to the reader. Specifically, what technology was used for sequencing, and what methods were used for assembly, as these have consequences on the quality and reliability of annotations. In general, the use of the short-read assemblies for the annotation of TR regions is concerning, and such issues have been consistently raised in the literature. The authors should comment on their choice to use short-read assemblies specifically, rather than other available datasets for naked mole rat that include long-read sequencing data as well. It is particularly concerning that one of the assemblies used is derived from DNA isolated from >1 animal. How the authors have dealt with the presence of heterozygosity in these assemblies should be outlined and discussed. It seems one key aspect of this manuscript is to promote the use of these data as a resource, users need to be made aware of what lengths were taken to ensure the best possible dataset was used and is being provided for the analysis conducted in this study and future studies wishing to use this resource.

Line 470: The authors list many scaffolds that are used for their analysis, but not detail is provided that explains/outlines how these were used to arrive at the data presented in Figure 1. For example the two scaffolds listed for TRG are not identical to one another. How did the authors choose which assembly to use? Did they assess whether differences reflecting sequencing, assembly errors, genuine polymorphism etc.?

Lines 483-484: Details on this should be reported in results. How many genes harbored multiple alleles, and how was the "same position" determined given that the authors are using scaffold, rather than full-locus assemblies?

Reviewer #2 (expert in $\gamma\delta$ T cells):

The naked mole rat (NMR) is a eusocial, subterranean rodent that is increasingly attracting the attention of the biomedical research community. Of interest to this work are its longevity, reduced incidence of tumors and the absence of age-associated thymus involution. The recently reported lack of NK cells is of particular interest, given the generally assumed importance of these cells for tumor control.

Sanchez Sanchez et al used databases and additionally collected data to analyze the (single) transcriptomes of "innate" T cells that could have taken over the function of NK cells. They were able to detect only a very small MAIT cell population and a lack of iNKT cells and discovered a new NK cell-like $\gamma\delta$ T cell population. This population shares an invariant g-chain and a limited TCR δ repertoire with V(D)J genes sharing short homology repeats (microhomology domains) typical for invariant $\gamma\delta$ TCR. The CDR3 of the NMR invariant is identical in length and shows physicochemical similarities with those g-chains of fetal human GV8 positive human fetal gd T cells and fetus-derived mouse (V γ 5/6) $\gamma\delta$ T cells, despite differential GV usage. In contrast to the above-mentioned invariant $\gamma\delta$ T cells, the NMR invariant $\gamma\delta$ T cells are also produced in the adult thymus. Another $\gamma\delta$ T cell population but with a variable TCR repertoire was also identified and this shows a gene expression profile associated with type 3 immune response. In addition, the work contains other interesting information: e.g. annotation of TCR(TR) loci, interspecies comparison of butyrophilin family members, analysis of the $\alpha\beta$ TCR repertoire and (re)definition of T cell

populations based on gene expression patterns.

I think the finding that $\gamma\delta$ T cells can be used as a substitute for a complete population of innate lymphocytes (NK cells) is an interesting and truly new one. NMR is not an "ordinary" model organism and it is (almost) impossible to collect functional data on the immune response. However, this and other studies in recent years show that single-cell transcriptomics (and certainly other single cell-omics in the future) allow by direct comparison of animals with a peculiar life style or phylogenetic origin to define critical features of immune system development and function. Basically, the paper is clearly structured and well written. However, parts of it, the genome analysis in particular, are difficult to understand for non-specialists. In addition, there are points that should be clarified. Here are my detailed comments.

The first section contains the NMR TCR (TR) gene annotation, which is essential for the study. However, Figure 1 is a copy-paste of images from the IMGT website, which was already published in 2021 (<https://www.imgt.org/IMGTrepertoire/LocusGenes/> see the respective Locus representations). It would be appropriate to point this out, especially since the images on the website are better readable and since much more critical information is provided in the links. Instead using these figures for Figure 1, I would recommend creating a new image that contains information that is particularly important for this work, which deals (almost exclusively) with TRG and TRD, and to highlight the relevant genes. It should also be emphasized that the distinction between "functional" on the one hand and "ORF" and "pseudogene" on the other is of very limited significance for statements about functionality.

A surprising finding is the use of the pseudogene TRGC1 by GV3-1. Please briefly describe exactly how this was tested. Was there a sequencing of entire TCRg chain transcripts or is this statement based on that parts of the genomic sequence were found in the transcripts? Please also describe which J and C genes are used by the GV3-2 TCRs.

Figure 2A. I'm not sure if I understood this figure and lines 141-155.

My understanding of the figure is: NMR GV1 branches out with the mouse GV5 (the DETC GV) and further away with the human GV1 containing the invariant GV8. Mouse GV4 stands on its own (I can't read the gene name). NMR GV2 (group) includes human GV9 and NMR GV2. The GV3 group of the mouse (GV5 gene) stands on its own. NMR-GV3 group and genes branch out with the human GV10 gene. The TGV2 group of mice (TRGV4 gene?) branches out with the human or with the NMR-GV4 group (human TRGV10 gene). GV4 of the NMR stands on its own (TRGV4:01, :02,03), as is the case with the mouse GV1 (TRGV1:01, 2:01, 3:01). In any case, please rewrite the relevant text and provide better legend and a better to read figure. Also, it took me some time to learn that the (hard-to-read gene annotation reads from left to right and not from the center outwards (group/gene/gene species). Please also indicate what kind of sequences were compared. V-exon only, protein or nucleotides?

In Figure 1S3. What type of sequences were analyzed ? Nucleotide sequence of genes, ORFs or protein sequences ? In the case of protein sequences, the whole protein or only protein domains. Especially with the BTN3 family, this can lead to different results. In addition, I doubt that the closest relative of the mouse's Skint1 is BTNL1. Mouse Skint1 is a member of a gene family and has homologues (SKINT1L) in many mammalian species. Macaques, for example, have a functional gene. See also: Origin and evolution of dendritic epidermal T cells. Sutoh Y et al. Front Immunol. 2018. The SKINT1-like gene is inactivated in hominoids, but not in all primate species: implications for the origin of dendritic epidermal T cells. Mohamed RH et al. PLoS One. 2015.

Fig. 5. Four abTCR-dominated clusters (0-3), including one CD8ab cluster (3) and two gd-T cell clusters (4 and 5) were identified. Cluster 4 genes are typical of type 3 T cells (curiously without CD4). Cluster 5 expresses CTL/NK-defining genes (Fig. 5C). When reviewing the xls files, I found not information on cytokines. Does this have a physiological significance or is it due to technical problems? Please explain this briefly. In any case, if there is no lymphokine signal (at least not for lymphokines abbreviated with IL-) mentioning that IL-17 is not expressed in cluster 4 is irrelevant. FoxP3, is shown in Fig. 6 as being expressed in individual cells, but not listed corresponding table.

Is this due to the low frequency of FoxP3-positive cells? If yes, it might help to display in Fig. 5 also cytokines in the single-cell plot (similar as in Fig. 6 for FoxP3). A low frequency of cytokine producing cells would be expected under steady state conditions, while surface molecules e.g. CD8b are expressed constitutively by all MHC I restricted cells.

RESPONSE TO REVIEWERS' COMMENTS

Reviewer #1

In this manuscript, Sanchez et al. have conducted an analysis of the T cell receptor (TCR) repertoire in the naked mole-rat, including: (1) germline gene annotations for each TCR locus in this species, (2) analysis of alpha, beta, delta and gamma TCR transcripts, and (3) focused analysis of single-cell RNA transcriptomes associated with invariant gamma/delta natural killer-like effector T cells. The motivation for this analysis is driven by a need to better understand T cell mediated immunity in this unique species. Overall, the manuscript is well-written and easy to follow. Detailed concerns/comments are provided below.

We thank the reviewer for this summary and the appreciation of the quality of the writing of the paper.

Below we provide a detailed point-to-point to the concerns and comments that are raised.

1.

Line 62: should be "gene".

This is adapted in the revised manuscript.

2.

Line 67: "at 'the' protein level"

This is adapted in the revised manuscript.

3.

Line 92: It is not exactly clear that this study "shows" that these cells have cytotoxic effector functions, but maybe more specifically that these cells have a specific marker/transcriptomic phenotype. The wording should be tempered here.

We followed the suggestion of the reviewer and have used now the wording 'phenotype', rather than 'function', in the revised manuscript (Introduction, page 4).

4.

Line 98: The legend for Figure 1 does not sufficiently explain features of the figure. Much more detail is needed. In the figures/maps themselves, it is not clear what scaffolds are being used for which parts of the maps, yet the authors say that more than one scaffold is used. And the methods claims that two different assemblies are used even, but there is no indication here what assemblies are represented in these maps. What do the double slanted lines represent. Are these "gaps" in the assembly/scaffold? What are the black arrows under assembly in panel B? Why are the genes above those arrows named differently? Are they TRA genes?

We agree with the reviewer that more details can be provided. Importantly, following the main comment of the reviewer, we have revised completely Figure 1. We refer to reviewer #1, points 30-32 for further details.

5.

Line 110: Instead of the use of "borne", the authors should consider using terms common to the genomics field to explain this. For one, it hasn't yet been defined by this line in the manuscript, but it seems the authors mean the nearest 5' non-TR flanking gene in the assembly/scaffold. To improve readability/comprehension, the authors should probably just say that instead of using uncommon jargon.

The term 'borne' is a standardized term, which is part of the IMGT ONTOLOGY.

Following the remark of the reviewer, we explain now the term 'borne' as follows in the revised manuscript, so that it is clear to the reader what the term means, and added a link and a reference (Results section, page 5):

"The delimitation of the locus was based on the investigation of the "IMGT bornes", which are coding genes, other than TR, that remain conserved across different species and are located upstream of the first or last gene of the locus, respectively (<http://www.imgt.org/IMGTindex/IMGTborne.php>; Manso et al 2022 Nucleic Acids

Research doi.org/10.1093/nar/gkab1136). Specifically, the 5' borne (OR10G3) and 3' borne (DAD1) ...”

In addition, we added as well a reference in the legend of Figure 1 where we use now also the term ‘borne’.

6.

Line 110: Check Nat Comm conventions whether specific gene names should be italicized.

Indeed, gene names are italicized in Nat Commun articles. We thank the reviewer for this remark. Therefore, we consistently now italicized gene names in the revised manuscript.

7.

Line 127: The authors mention that some genes have “provisional” names. It’s not clear what is mean by this, and based/on compared to what. In addition, this is confusing as the maps shown in Figure 1 look to be claiming that all names are “provisional”. Not clear what this means, how it was decided upon, or how to interpret.

Because of the use of the chromosomal assemblies (following the important suggestion of this reviewer, see also reviewer #1 points 30-32), there is no provisional nomenclature anymore. The naked mole-rat TR loci have now their definitive nomenclature and NCBI accession numbers have been obtained.

8.

Line 131: Not clear what ‘invariant’ means in this section. Can you define what is meant by invariant when referring to germline genes? And why it is relevant to the phylogenetic analysis in this section? Is this a term for germline genes that contribute to ‘invariant’ TCRs? If so this should be stated, and also important to clarify whether there are TRG/D genes that are not ‘invariant’.

We thank the reviewer for this remark. We realized that in the context of describing germline genes the term ‘invariant TRGV or TRDV’ can indeed be confusing.

We have rewritten this section in the revised manuscript to explain invariant $\gamma\delta$ TCRs present in human and mouse, the TRGV and TRDV genes used in these invariant $\gamma\delta$ TCRs and how these relate to some of the TRGV and TRDV genes in the naked mole-rat (Results section, page 6).

9.

Lines 136-138: Why are those genes exceptions? And how were exceptions decided upon?

We explain this now in the revised version of the manuscript (Results section, page 6):

‘The subgroup classification of the naked mole-rat’s TR loci is primarily based on human subgroups, with a few exceptions such as TRBV1, TRDV4, and TRDV5. These exceptions arise from their significantly low genetic identity percentages, which hinder correspondence with human and mouse subgroups’.

10.

Line 147: How is “high homology” defined?

We followed the remark of the reviewer and have rewritten this part using now the term ‘identity’, instead of ‘homology’, which is the correct term in this sentence (Results, page 6):

‘Interestingly, the naked mole-rat possess TRGV (TRGV2) and TRDV (TRDV2) genes that show high identity with the human TRDV2 and TRGV9 genes respectively (Fig. 2), while the mouse that does not show significant identity with these human genes.’

11.

Figure 2/S1: Legend needs citations for MEGA, MAFFT, etc.

We have now updated the citations in the legend of both figures. A more detailed explanation of different methodologies can be found as well in Methods section.

12.

Lines 150-151: Can the authors provide support for this claim beyond just the assembly data? Were these genes also not observed in the expressed TCR repertoire?

Is regarding: ‘Of note, both the naked mole-rat TRGV9 and TRDV2 genes are pseudogenes (Fig. 1A-B).’

We have generated now a new Table (Supplementary Table 2) in the revised version of the manuscript (also provided below) that contains the number of CDR3 reads per TRDV and TRGV gene. We did not detect any CDR3 read that contained either the TRGV2 or the TRDV2 pseudogenes from the naked mole-rat. Overall, assembly annotation data correlated with the nature of the expressed TCR repertoire, with the exception of detection of TRGC1-containing sequences (a constant gene identified as pseudogene) that was already discussed in the manuscript (lines 422 to 429 of the original submission). This new supplementary Table 2 illustrates as well the data displayed in original Figure 3B (now Figure 4B) that contains boxplots of TRGV (top) and TRDV (bottom) use of neonate blood (n=3) and adult blood and spleen (n=7).

13.

Line 155: This claim should be tempered. Fair to say, no clear orthologue was observed in the assemblies you’ve analyzed, but not extrapolating to the species as a whole might be a leap.

Is regarding: ‘For the mouse invariant TRGV genes (TRGV5 and TRGV6) no clear orthologue exists in the naked mole-rat (Fig. 2A).’

Besides the available contig/scaffold assemblies, we have verified now also the chromosomal assemblies. No clear orthologues were found for mouse TRGV5 and mouse TRGV6 in any of these assemblies. Thus we think that we can conclude that ‘For the mouse invariant TRGV genes (TRGV5 and TRGV6) no clear orthologue exists in the naked mole-rat’ and thus we would rather keep this sentence.

14.

Lines 156-162: This seems out of place given the header of the results section. Its relevance should be better motivated, perhaps written into its own section.

We have extended the motivation of the analysis of BTN(L) members in the naked mole-rat and have, as suggested by the reviewer, put the description of the results of this analysis now in a separate section in the revised version of the manuscript (Results section, page 7).

15.

Line 164: At the beginning of this section, the authors should state which genes from the annotations in Figure 1 are supported at nucleotide level from TCR transcript reads from the animals analyzed.

Given that the primary focus of our article is the $\gamma\delta$ T cell compartment, we have incorporated a detailed table (see also our answer to point 12 of reviewer #1) in the revised version of the manuscript which classifies reads based on the TRGV and TRDV genes they contain (Supplementary Table 2, also provided below) thus allowing a comparison at the level of TCR transcription with the genes depicted in the loci of Figure 1. Regarding TRAV and TRBV usage, we refer now more explicitly to Figure S2 (original submission) where boxplots are provided of TRAV and TRBV usage from the distinct repertoires of neonatal blood samples and adult blood/spleen samples.

16.

Line 172 (Figure 3B): Why did the authors combine adult blood and spleen for this analysis?

We aimed to highlight the preferential usage of specific TRDV genes in neonates compared to other stages. To enhance the statistical power and increase our sample size (n), we merged the two adult datasets. We would like to perform new bulk TCR-seq experiments to increase n in the different age and tissue groups.

17.

Line 173: Believe it should be “tendency”.

This is adapted.

18.

Lines 166-168: The number of animals studied, and their origins should be stated here in text (rather than make reader go methods or figure legends). Are they unrelated animals, littermates, etc.?

We have provided in the revised version more information regarding the number and origin (littermates) in the Results section. Further details regarding which animal is from which litter is now provided in the Methods section and supplementary Table 1.

19.

Figure 3B: What is the basis for including the genes that are plotted. Usage levels? Is the reader to assume that all other genes have no usage? It would also be useful to comment on the fact that 3 of the 4 genes shown are annotated as ORF, rather than functional. What is the basis for that?

We have now included a comment in the corresponding legend stating that genes shown correspond to all the detected transcripts in our TCR sequencing data. We also refer in the main text from the revised version to the Supplementary Table 2 that provides (with numerical values) the levels of each detected TRGV and TRDV gene.

The ORF TRGV and TRDV genes in Figure 3B (original submission) have been annotated as ORF, as their coding region has an open reading frame without stop codons but shows alterations in the recombination signals, deletions of more than 3 amino acids in the V-REGION and/or changes of conserved amino acids that might lead to incorrect protein folding. It is important to note that the annotation of a gene as ORF does not imply that it cannot contribute to a functional TCR. It rather suggests that there are alternations that cannot ensure its functionality

(<https://www.imgt.org/IMGTScientificChart/SequenceDescription/IMGTfunctionality.html>). However, the definition of functionality within IMGT is a dynamic process. It is continuously evolving in response to new experimental data, particularly when applied to new species, where these changes may be unique and characteristic of a specific species. We explain now in more detail the designation of F (functional), ORF (open reading frame) and P (pseudogene) in the revised version of the manuscript, and explain that the assignment of ORF (versus F) is a dynamic process.

20.

Line 167: What is the rationale for studying these tissues?

The primary factor motivating the choice of spleen and blood for this first investigation of the TCR repertoire in the naked mole-rat is the feasibility of obtaining these tissues at various developmental timepoints, including neonates. The naked mole-rat represents an unconventional animal model, and the procedures for collecting organs and optimizing these collection methods have not been comprehensively documented. We think that investigating immune cells associated with other tissues than spleen and blood in the naked mole-rat is of interest and should be a topic in future studies but is out of scope of the current study.

21.

Line 177-179: The authors should dissect this claim. What is it based on? If the authors have simply run a statistical test in each group separately and then made this assessment based on individual P values, may not be appropriate. Authors should be more explicit.

Is regarding: ‘Differences in junctional diversity, CDR3 nucleotide length and NDN size between neonates and adults were more prominent in TRDV1-6-containing sequences than TRDV1-4 (Fig. 3C).’

As highlighted by the reviewer, the p-values in the figure correspond to comparisons that are not directly connected to the statement. Furthermore, the presence of non-significant differences in some of these comparisons could potentially lead to confusion for the reader. In this last case we think that this is explained by the low number of samples compared, and the statistical test used (Kruskal-Wallis test). Because the current p values depicted in Fig. 3C (original submitted version; now Fig. 4C), we removed these and we adapted the corresponding text in the revised manuscript (Results section, page 8). We would like to increase n in the different groups to enhance the statistical power.

22.

Line 190: Metric used for “sharing” (F value) should be stated here in results. “Repertoire sharing” is vague.

Is regarding: ‘Consistent with reduced junctional diversity during the neonatal period in the naked mole-rat, the levels of TRB, TRG and TRD repertoire sharing within neonatal samples were higher compared to their adult counterparts (Fig. 3F).’

We followed the request of the reviewer and we state now in the revised version of the manuscript the metric for sharing between two samples, that is the F value, also in the Results section (besides in the Figure legend of Figure 3F (new Figure 4F)). Furthermore, we provide now more details regarding the calculation of the F value in the Methods section.

23.

Figure 4 legend: typo? “TRG5-3”

Indeed, this should be TRGJ5-3 and has been adapted in the manuscript.

24.

Figure 5D: Could the authors comment on low detection of TRGV3-1 in cluster 4? It seems that the frequency of detected gamma TCRs is low relative to delta? The inverse seems to be somewhat true for clusters 3 and 5. How have the authors verified they haven’t artificially missed some unidentified TCR chains?

The reason for the observed differences in TRG and TRD detection between Clusters 4 and 5 can be related to the fact the different types of $\gamma\delta$ T cells can express different $\gamma\delta$ TCR levels. In the mouse, the $V\gamma5+$ $\gamma\delta$ subset has been shown to express high TCR levels compared to other $\gamma\delta$ subsets (e.g., Sumaria 2011 J Exp Med doi/10.1084/jem.20101824; Van hede 2017 PNAS doi/10.1073/pnas.1712883114), while in human the $V\delta1$ subset express higher TCR levels than the $V\delta2$ subset (e.g., Yokobori et al 2009 Clin Exp Immunol 10.1111/j.1365-2249.2009.03974.x; Jagannathan et al 2014 Sci Transl Med 0.1126/scitranslmed.3009793). We therefore propose that the two $\gamma\delta$ T cell clusters detected in the naked mole-rat, that are associated with different effector profiles, show also such differences in TCR expression levels which is then reflected in different levels of TRG and TRD CDR3 detection by the TRUST4 package based on single-cell gene expression.

The transcriptomic profile of Cluster 3 (as depicted in Figure 5C, original submission) and the TCR sequences identified by TRUST4 (as illustrated in Figure 5B, original submission) suggests that Cluster 3 predominantly comprises CD8 TCR $\alpha\beta$ cells. Despite this conclusion, as the reviewer pointed out, we are able to detect CDR3 γ sequences in this cluster. This seemingly contradictory result can be explained by the fact that TCR $\alpha\beta$ T cells have the capacity to express transcripts of rearranged TRG sequences (Sherwood et al. 2011 Sci Transl Med 10.1126/scitranslmed.3002536). In line with this, almost no CDR3 δ sequences can be detected in this TCR $\alpha\beta$ cluster, a result that is expected as TRD genes are interspersed with TRA genes in the same loci (as illustrated in Figure 1): the expression of a TRA chain excludes the expression of a TRD chain.

25.

Line 297: number of animals should be stated.

In line 297 (original submission), the number of animals used in this analysis was already specified: ‘*In order to extend our findings in the blood to the spleen compartment, we re-analyzed a public available dataset from spleens of **four** young-adult animals (23 to 25 months old)*’

26.

Line 299: Figure 6B, no x axis labels. Looks as though it should be clusters from panel A.

We apologize for missing the x axis label in Figure 6B (original submission). This x axis is indeed referring to the clusters identified in Figure 6A (original submission). We have now added the cluster numbers on the x axis of the figure panel shown in Figure 6B (now Figure 7B in the revised version) and mention in the corresponding legend that the numbers correspond to the clusters identified in Figure 6A (now Figure 7A).

27.

Lines 325: number of animals included in analysis should be stated.

We have now included in this sentence the number of animals in both age groups (young-adult and middle-aged).

28.

Figure S6C: Potential typo. Should it be “other” rather than “ather”?

Indeed, this should be ‘other’, thank you for pointing this out. The legend key of Figure S6C has been updated accordingly.

29.

Line 427: Typo. “TRCG”

Indeed, this should be TRGC and has been adapted in the manuscript.

30.

Line 459: The authors should better outline the source of the scaffold assemblies used so it is clear to the reader. Specifically, what technology was used for sequencing, and what methods were used for assembly, as these have consequences on the quality and reliability of annotations. In generally, the use of the short-read assemblies for the annotation of TR regions is concerning, and such issues have been consistently raised in the literature. The authors should comment on their choice to use short-read assemblies specifically, rather than other available datasets for naked mole rat that include long-read sequencing data as well. It is particularly concerning that one of the assemblies used is derived from DNA isolated from >1 animal. How the authors have dealt with the presence of heterozygosity in these assemblies should be outlined and discussed. It seems one key aspect of this manuscript is to promote the use of these data as a resource, users need to be made aware of what lengths were taken to ensure the best possible dataset was used and is being provided for the analysis conducted in this study and future studies wishing to use this resource.

In our prior work, we relied on scaffold-based assemblies generated from BGISEQ-500 sequencing, specifically the Heter_glaber.v1.7_hic_pac62 and the HetGla_female_1.063 assemblies, as they were the sole available assemblies in 2021 on NCBI for the naked mole-rat. These assemblies served as an expedient means for the temporary annotation of the TR loci in the naked mole-rat, as well as for subsequent analyses of TR expression data.

Importantly, we have now completed the full annotation of all four TR loci using the latest complete chromosome haploid assemblies, which were derived from long-read sequencing, aligning with the request of the reviewer. This accomplishment has led to the establishment of the definitive nomenclature for the naked mole-rat TR loci which is now presented in Figure 1 of the revised manuscript. This Figure 1 is also provided below.

Accession numbers have been obtained at the NCBI:

TRAD locus maternal BK064754

TRG locus paternal BK064755

TRG locus maternal BK064756

TRB locus paternal BK064758

TRB locus maternal BK064759

Notably, this final updated annotation has not resulted in substantial changes to the provisional nomenclature. Consequently, all analyses related to TR expression data in the naked mole-rat remain valid and have been further confirmed.

As evident from the new Figure 1, all the loci from different assemblies exhibit a consistent core set of genes. The order of the genes remains the same and no changes have been found to genes functionality. With the exception of only one new pseudogene, the TRAV23-6, no novel genes were detected that had not been previously identified in the scaffolds. However, it is noteworthy that polymorphism in the coding region of some genes were detected and annotated, enriching the NMR TR gene dataset with novel alleles. Of note, maternal and paternal haplotypes annotated for TRB and TRG loci did not show differences in the presence/absence or order of TR genes. Differences detected between the scaffold assemblies and the long-read sequences was the absence of certain genes in the chromosomal assemblies. The detailed list is given below. As noticed, those gene appeared after or between gaps. In addition, after/before each of them a highly identical gene (gene belonging to the same subgroup) is present. For instance, in the TRAD locus, TRAV12-1 was not found in any of the chromosomal assemblies. This gene is detected after a gap, followed by TRAV12-2, which belongs to the same subgroup. Our hypothesis is that due to the gaps, an 'artificial duplication' of TRAV12-2 has occurred, leading to the creation of the TRAV12-1.

Similar cases are observed as follows:

TRB: TRBV18-2, TRBV19-2 and TRBV24-1

TRG: TRGC2S1

TRAD: TRAV12-1, TRDV1-3, TRAV5-2, TRAV22-5, TRAV23-5, TRDVC, TRAV25-2, TRAV8-4, TRAV8-5, TRAV8-7, TRAV38-1, TRAV38S1, TRAV38S2, TRAV35S2, TRAV34S1 and TRAV 35S3.

As mentioned above, the final updated annotation has not resulted in any change related to TR expression data in the naked mole-rat and thus the TR expression analysis and associated conclusions remain valid.

A**Naked mole-rat (*Heterocephalus glaber*) TRG locus on chromosome 1****B****Naked mole-rat (*Heterocephalus glaber*) TRB locus on chromosome 22****C****Naked mole-rat (*Heterocephalus glaber*) TRA/TRD locus on chromosome 2**
Figure 1. Locus representation of the *Heterocephalus glaber* TR loci of the assembly Naked mole-rat maternal (GCA_944319715.1). The figures display the position, nomenclature and functionality of all TR genes according to the IMGT nomenclature⁶⁸. A dotted line ... indicates the distance in kb between the locus and its IMGT borne⁶⁴ (5' borne and the most 5' gene in the locus and/or 3' borne and the most 3' gene in the locus) on the top and bottom lines, respectively. These distances are not represented at scale and are not included in the numbers displayed at the right ends of these two lines. Finally, the black arrows indicate an inverse transcriptional orientation in the locus while the blue arrows indicate the position in the TRG and TRD locus of V genes that are highlighted in the current study.

31.

Line 470: The authors list many scaffolds that are used for their analysis, but not detail is provided that explains/outlines how these were used to arrive at the data presented in Figure 1. For example the two scaffolds listed for TRG are not identical to one another. How did the authors choose which assembly to use? Did they assess whether differences reflecting sequencing, assembly errors, genuine polymorphism etc.?

Following the important remark of this reviewer (see also point 30), the annotation based on long-read sequence assemblies (chromosome level) has been completed. The results of the chromosomal assemblies' annotation align with our previous findings and validate them as only minor differences were identified. Therefore, new maps have been generated based on the new assemblies, with no gaps and providing the definitive nomenclature of the naked mole-rat TRG, TRAD and TRB loci with NCBI accession numbers (see also the reply to point 30).

32.

Lines 483-484: Details on this should be reported in results. How many genes harbored multiple alleles, and how was the "same position" determined given that the authors are using scaffold, rather than full-locus assemblies?

Is about: 'Additionally, new alleles were annotated when V-REGIONS in different assemblies had an identity score of approximately 99% and were positioned at the same position within the locus. ...'

In our research, although we initially relied on scaffold-based assemblies to annotate the TR loci, as the chromosomal ones were not yet available in 2021, we performed a detailed localization process that relied on mapping and aligning existing gene sequences to the scaffold sequences to achieve the most accurate representation. In cases where the position or orientation was not clear, we used provisional nomenclature. These initial results regarding positions and gene orders are now confirmed and validated by the chromosomal annotation (see our reply to point 30).

The consistency between scaffold-based positions and chromosome-based positions provides strong support and validation of our methodology and our earlier findings and eliminates any concerns about the gene positions.

We have modified the corresponding text in the Methods section (page 17) accordingly:

'Additionally, new alleles were annotated when the coding part of each gene (V-REGION, J-REGION, D-REGION or C exons) had an identity score of approximately 99% in different assemblies.'

Reviewer #2

The naked mole rat (NMR) is a eusocial, subterranean rodent that is increasingly attracting the attention of the biomedical research community. Of interest to this work are its longevity, reduced incidence of tumors and the absence of age-associated thymus involution. The recently reported lack of NK cells is of particular interest, given the generally assumed importance of these cells for tumor control.

Sanchez Sanchez et al used databases and additionally collected data to analyze the (single) transcriptomes of "innate" T cells that could have taken over the function of NK cells. They were able to detect only a very small MAIT cell population and a lack of iNKT cells and discovered a new NK cell-like $\gamma\delta$ T cell population. This population shares an invariant g-chain and a limited TCR δ repertoire with V(D)J genes sharing short homology repeats (microhomology domains) typical for invariant $\gamma\delta$ TCR. The CDR3 of the NMR invariant is identical in length and shows physicochemical similarities with those g-chains of fetal human GV8 positive human fetal gd T cells and fetus-derived mouse (V γ 5/6) $\gamma\delta$ T cells, despite differential GV usage. In contrast to the above-mentioned invariant $\gamma\delta$ T cells, the NMR invariant $\gamma\delta$ T cells are also produced in the adult thymus. Another $\gamma\delta$ T cell population but with a variable TCR repertoire was also identified and this shows a gene expression profile associated with type 3 immune response. In addition, the work contains other interesting information: e.g. annotation of TCR(TR) loci, interspecies comparison of butyrophilin family members, analysis of the $\alpha\beta$ TCR repertoire and (re)definition of T cell populations based on gene expression patterns.

I think the finding that $\gamma\delta$ T cells can be used as a substitute for a complete population of innate lymphocytes (NK cells) is an interesting and truly new one. NMR is not an "ordinary" model organism and it is (almost) impossible to collect functional data on the immune response. However, this and other studies in recent years show that single-cell transcriptomics (and certainly other single cell-omics in the future) allow by direct comparison of animals with a peculiar life style or phylogenetic origin to define critical features of immune system development and function. Basically, the paper is clearly structured and well written. However, parts of it, the genome analysis in particular, are difficult to understand for non-specialists. In addition, there are points that should be clarified. Here are my detailed comments.

We are grateful to this reviewer for this very nice overview and highlighting the strengths of our paper.

The comment related to the comparison of the invariant NMR TRGV4-2 sequence with invariant CDR3 sequences in mouse and human triggered us to put the associated figure showing these intriguing similarities in a main figure (new Figure 5C) rather than in a supplemental figure as it was in the original submission.

As detailed below, based on the comments of both reviewers, we have significantly improved the manuscript, especially regarding the genome assembly/annotation.

1a.

The first section contains the NMR TCR (TR) gene annotation, which is essential for the study. However, Figure 1 is a copy-paste of images from the IMGT website, which was already published in 2021 (<https://www.imgt.org/IMGTrepertoire/LocusGenes/> see the respective Locus representations). It would be appropriate to point this out, especially since the images on the website are better readable and since much more critical information is provided in the links. Instead using these figures for Figure 1, I would recommend creating a new image that contains information that is particularly important for this work, which deals (almost exclusively) with TRG and TRD, and to highlight the relevant genes.

In our prior work, we relied on scaffold-based assemblies generated from BGISEQ-500 sequencing, specifically the Heter_glaber.v1.7_hic_pac62 and the HetGla_female_1.063 assemblies, as they were the sole available assemblies in 2021 on NCBI for the naked mole-rat. These assemblies served as an expedient means for the temporary annotation of the TR loci in the naked mole-rat, which were indeed published in 2021 on the IMGT website, as well as for subsequent analyses of TR expression data.

The updated annotation of the long-read sequences of the chromosome assemblies is now accompanied with the updated and definitive TR locus representation in the new Figure 1 (see also our replies to points 30/31 of reviewer #1). These new representations for each TR locus and assembly will be published for the first time in this article. In response to the reviewer's recommendation, we have updated the figures, as well as the legends and labels, to be more

detailed and descriptive, to enhance the clarity and relevance of our publication and give the reader a better understanding of our findings. Furthermore, we indicate now with arrows in Figure 1 the relevant genes that are further studied in detail (at the expression level) in the remaining parts of the article.

1b.

It should also be emphasized that the distinction between “functional” on the one hand and “ORF” and “pseudogene” on the other is of very limited significance for statements about functionality.

Following this comment, we have now included the text below in the Results section (page 5) of the revised manuscript:

‘A gene is assigned ORF (open reading frame), rather than F (functional) when the coding region has an open reading frame without stop codons (like in a F) but shows (unlike the F designation) gene alterations in the recombination signals (compared to the canonical recombination signal sequences), deletions of more than 3 amino acids in the V-REGION and/or changes of conserved amino acids that might lead to incorrect protein folding. It is important to note that the annotation of a gene as ORF (versus F) is a dynamic process, particularly when applied to new species, where these changes may be unique and characteristic of a specific species. A gene is assigned P (pseudogene) when the coding region has stop codon(s) and/or frameshift mutation(s) and/or a mutation affecting the initiation codon.’

2.

A surprising finding is the use of the pseudogene TRGC1 by GV3-1. Please briefly describe exactly how this was tested. Was there a sequencing of entire TCRg chain transcripts or is this statement based on that parts of the genomic sequence were found in the transcripts? Please also describe which J and C genes are used by the GV3-2 TCRs.

The identification of sequences containing TRGV3-1 and TRGC1 genes was achieved with TRUST4 analysis, utilizing the raw single-cell sequencing files. TRUST4 is designed to extract transcripts that encompass V(D)J genes and, through a scoring system, determine the specific gene being utilized. The output from TRUST4 provides detailed information about the nucleotides within the transcript sequences that correspond to each gene. This feature simplifies direct comparisons with the annotation database used as a reference for the CDR3-sequence extraction protocol. It is important to note that, in some cases, due to transcript length, TRUST4 may not yield enough information for a confident prediction of the TRGC gene used in the sequence. Nonetheless, such sequences are retained in the analysis, as the primary goal for the software is to reconstruct the CDR3 region, and if possible, provide the constant gene.

All the identified sequences containing the TRGC1 gene are consistently paired with a TRGJ1-2 gene, aligning with the cassette structure illustrated in Figure 1. Likewise, all sequences containing TRGC5 are consistently paired with TRGJ5-3, affirming the robustness of our TCR loci annotation and the accuracy of the CDR3 data extracted using TRUST4.

3.

Figure 2A. I'm not sure if I understood this figure and lines 141-155.

My understanding of the figure is: NMR GV1 branches out with the mouse GV5 (the DETC GV) and further away with the human GV1 containing the invariant GV8. Mouse GV4 stands on its own (I can't read the gene name). NMR GV2 (group) includes human GV9 and NMR GV2. The GV3 group of the mouse (GV5 gene) stands on its own. NMR-GV3 group and genes branch out with the human GV10 gene. The TGV2 group of mice (TRGV4 gene?) branches out with the human or with the NMR-GV4 group (human TRGV10 gene). GV4 of the NMR stands on its own (TRGV4:01,;02,03), as is the case with the mouse GV1 (TRGV1:01, 2:01, 3:01). In any case, please rewrite the relevant text and provide better legend and a better to read figure. Also, it took me some time to learn that the (hard-to-read gene annotation reads from left to right and not from the center outwards (group/gene/gene species). Please also indicate what kind of sequences were compared. V-exon only, protein or nucleotides?

We understand the confusion of the reviewer and we would like to thank them for this feedback. Some of the mouse subgroup names have the same name of particular mouse TRGV genes that are used to identify particular mouse $\gamma\delta$ T cells subsets. To avoid this and thus increase clarity of Figure 2A, we have removed the 'subgroup nomenclature' (Figure 2 is also provided below). Furthermore, we have improved resolution of the figures and used abbreviations for the species annotation that have resulted in a figure that is much better readable.

The phylogenetic trees of the TRV genes is based on nucleotide sequence alignments. We have indicated this now in the legend of Figure 2, in the Results section (page 6) and in the Methods section (page 18).

Figure 2. Unrooted phylogenetic trees of TRGV (A) and TRDV (B) genes in *Homo sapiens*, *Mus musculus*, and *Heterocephalus glaber*. The trees were constructed using the Maximum Likelihood method in MEGA software⁷⁴ The trees are based on nucleotide sequence alignments, generated with representatives of V-REGION sequences from each subgroup using MAFFT⁷⁵. Visualization of the trees was carried out using iTOL v6⁷⁶, with distinct colors representing the different human subgroups and their orthologs. Species abbreviations are as follows: HMN, *Homo sapiens*; MMU, *Mus musculus*; NMR, *Heterocephalus glaber*.

4a.

In Figure 1S3. What type of sequences were analyzed ? Nucleotide sequence of genes, ORFs or protein sequences ? In the case of protein sequences, the whole protein or only protein domains. Especially with the BTN3 family, this can lead to different results.

We thank the reviewer for this remark. This was indeed not clear from the original submission. The analysis was based on protein sequences. We have detailed this analysis now in the legend of the phylogenetic tree of the BTN(L) sequences of the naked mole-rat, human and mouse:

‘Maximum Likelihood tree of BTN(L) proteins was constructed utilizing MEGA software, based on amino acid MAFFT sequence alignments of the BTN(L) and SKINT proteins obtained from NCBI. We used proteins, as there was enough information to reconstruct the phylogenetic tree. A single representative sequence from each protein family was selected for inclusion in the analysis. The tree was visualized with iTOL v6, and manually assigned colors indicate orthologous proteins, determined using the NCBI ortholog database. For example, the BTNL9 proteins from HMN, MMU, and NMR share a branch (colored orange) as orthologs. Species abbreviations are as follows: HMN, *Homo sapiens*; MMU, *Mus musculus*; NMR, *Heterocephalus glaber*.’

Furthermore, we provide now this revised version of the BTN(L) phylogenetic tree as a new Figure 3 (provided below), following the introduction in the revised version of the manuscript of a separate paragraph in the results section on this topic: see point 14 of reviewer #1. We provide also more details in the Methods section (page 18) regarding the phylogenetic tree construction.

4b.

In addition, I doubt that the closest relative of the mouse's Skint1 is BTNL1. Mouse Skint1 is a member of a gene family and has homologues (SKINT1L) in many mammalian species. Macaques, for example, have a functional gene. See also: Origin and evolution of dendritic epidermal T cells. Sutoh Y et al. Front Immunol. 2018. The SKINT1-like gene is inactivated in hominoids, but not in all primate species: implications for the origin of dendritic epidermal T cells. Mohamed RH et al. PLoS One. 2015.

In the original submission, no distinction was made between the different members of the mouse Skint family, leading to confusion. In the revised version of the figure, we have included all Skint family members (Figure 3 from revised manuscript; also included below). As the reviewer correctly pointed out, this analysis clearly shows that BTNL1 is not the closest relative to Skint1. The closest relatives are other members of the Skint family, such as Skint2, Skint 5 and Skint6. In contrast, mouse BTNL1 clusters together with mouse BTNL12.

Furthermore, we confirm that we did not find SKINT1L in the naked mole-rat. Human SKINT1L is not included in our phylogenetic tree, since our analysis is based on protein sequences (as explained now in more detail in the legend of revised version of the figure legend) and human SKINT1L is a pseudogene.

Figure 3. Unrooted phylogenetic trees of BTN(L) molecules from *Homo sapiens*, *Mus musculus*, and *Heterocephalus glaber* species. Maximum Likelihood tree of BTN(L) proteins was constructed utilizing MEGA software⁷⁴, based on amino acid MAFFT⁷⁵ sequence alignments of the BTN(L) and SKINT proteins obtained from NCBI. Protein sequences were used as there was enough information to reconstruct the phylogenetic tree. A single representative sequence from each protein family was selected for inclusion in the analysis. The tree was visualized with iTOL v6⁷⁶, and manually assigned colors indicate orthologous proteins, determined using the NCBI ortholog database. For example, the BTNL9 proteins from HMN, MMU, and NMR share a branch (colored orange) as orthologs. Species abbreviations are as follows: HMN, *Homo sapiens*; MMU, *Mus musculus*; NMR, *Heterocephalus glaber*.

5.

Fig. 5. Four abTCR-dominated clusters (0-3), including one CD8ab cluster (3) and two gd-T cell clusters (4 and 5) were identified. Cluster 4 genes are typical of type 3 T cells (curiously without CD4). Cluster 5 expresses CTL/NK-defining genes (Fig. 5C). When reviewing the xls files, I found not information on cytokines. Does this have a physiological significance or is it due to technical problems? Please explain this briefly. In any case, if there is no lymphokine signal (at least not for lymphokines abbreviated with IL-) mentioning that IL-17 is not expressed in cluster 4 is irrelevant. FoxP3, is shown in Fig. 6 as being expressed in individual cells, but not listed corresponding table. Is this due to the low frequency of FoxP3-positive cells? If yes, it might help to display in Fig. 5 also cytokines in the single-cell plot (similar as in Fig. 6 for FoxP3). A low frequency of cytokine producing cells would be expected under steady state conditions, while surface molecules e.g. CD8b are expressed constitutively by all MHC I restricted cells.

We appreciate the reviewer's feedback on this matter. During the course of our single-cell analysis, we did observe differences in sequencing depth between blood and spleen samples, which could potentially influence the transcripts detected in the cells. Despite this limitation, in both datasets, $\gamma\delta$ T cells displayed a prominent cytotoxic profile without expression of lymphokines associated with helper roles, such as IL-17 or IFN γ that are used to subdivide functional $\gamma\delta$ T cell subsets in mouse.

In all single-cell datasets, the expression of genes encoding lymphokine molecules in the T cell clusters (IL genes) were either absent or low (see Figure R1 below). Other cytokines and chemokines with helper roles could be detected but in general the expression was also low (Figure R1). We agree with the reviewer that steady-state conditions could play a role in these low detection levels. Therefore, in light of these considerations, we have decided to follow the reviewer's recommendation and remove the statement about IL-17 (Results page 10; Discussion page 16).

Finally, the absence of certain lymphokine genes and other functionality-related genes, such as FOXP3, in the xls file is due to our choice of a statistical test that does not include genes expressed in less than 15% of the cells within a single-cell cluster in the testing (explained in the Methods section).

Figure R1. UMAP plots representing a series of selected genes in single-cell gene expression data in blood T cells from the naked mole-rat.

Supplementary Table 2 (included in the revised version). Number of CDR3 reads divided according to TRDV and TRGV usage in RNA transcripts

sample id	total CDR3 reads	TRDV1-4	TRDV1-5	TRDV1-6	TRDV3	TRDV5		sample id	total CDR3 reads	TRGV1-1	TRGV3-1	TRGV3-2	TRGV4-2
CT_136	5545	1081	147	4164	117	36		CT_136	7758	46	12	80	7620
CT_N51	2620	1926	0	509	185	0		CT_N51	4600	0	0	0	4600
LN_134	987	521	5	460	1	0		LN_134	4264	13	14	21	4216
PB_001	1344	270	0	1074	0	0		PB_001	125	2	6	0	117
PB_134	8053	2631	6	5336	80	0		PB_134	7738	136	120	0	7482
PB_136	5303	1875	34	3367	27	0		PB_136	2473	31	41	13	2388
PB_N51	72	37	0	35	0	0		PB_N51	145	0	0	0	145
PB_N52	465	249	2	214	0	0		PB_N52	581	1	2	0	578
PB_N53	387	245	0	142	0	0		PB_N53	190	0	0	0	190
SP_134	13286	2803	0	10318	165	0		SP_137	5395	10	12	1	5372
SP_135	11836	4860	47	6929	0	0		SP_138	7058	25	12	13	7008
SP_136	5656	1552	21	4079	1	3		SP_139	5262	24	7	3	5228
SP_137	3022	1415	0	1588	7	12		TT_136	3615	13	30	34	3538
SP_138	6503	2467	9	4026	0	1		TT_N51	12147	882	0	378	10887
SP_139	1670	941	13	711	5	0		SP_135	1317	24	9	18	1266
TT_136	2173	548	94	1471	44	16		SP_136	13252	62	0	108	13082
TT_N51	14237	4052	0	10185	0	0							

REVIEWER COMMENTS

Reviewer #1 (expert in annotation and assembly of antigen receptor loci):

In this manuscript, Sanchez et al. have conducted an analysis of the T cell receptor (TCR) repertoire in the naked mole-rat, including: (1) germline gene annotations for each TCR locus in this species, (2) analysis of alpha, beta, delta and gamma TCR transcripts, and (3) focused analysis of single-cell RNA transcriptomes associated with invariant gamma/delta natural killer-like effector T cells. The motivation for this analysis is driven by a need to better understand T cell mediated immunity in this unique species. Overall, the manuscript is well-written and easy to follow. Detailed concerns/comments are provided below.

Line 62: should be "gene".

Line 67: "at 'the' protein level"

Line 92: It is not exactly clear that this study "shows" that these cells have cytotoxic effector functions, but maybe more specifically that these cells have a specific marker/transcriptomic phenotype. The wording should be tempered here.

Line 98: The legend for Figure 1 does not sufficiently explain features of the figure. Much more detail is needed. In the figures/maps themselves, it is not clear what scaffolds are being used for which parts of the maps, yet the authors say that more than one scaffold is used. And the methods claims that two different assemblies are used even, but there is no indication here what assemblies are represented in these maps. What do the double slanted lines represent. Are these "gaps" in the assembly/scaffolds? What are the black arrows under assembly in panel B? Why are the genes above those arrows named differently? Are they TRA genes?

Line 110: Instead of the use of "borne", the authors should consider using terms common to the genomics field to explain this. For one, it hasn't yet been defined by this line in the manuscript, but it seems the authors mean the nearest 5' non-TR flanking gene in the assembly/scaffold. To improve readability/comprehension, the authors should probably just say that instead of using uncommon jargon.

Line 110: Check Nat Comm conventions whether specific gene names should be italicized.

Line 127: The authors mention that some genes have "provisional" names. It's not clear what is meant by this, and based/on compared to what. In addition, this is confusing as the maps shown in Figure 1 look to be claiming that all names are "provisional". Not clear what this means, how it was decided upon, or how to interpret.

Line 131: Not clear what 'invariant' means in this section. Can you define what is meant by invariant when referring to germline genes? And why it is relevant to the phylogenetic analysis in this section? Is this a term for germline genes that contribute to 'invariant' TCRs? If so this should be stated, and also important to clarify whether there are TRG/D genes that are not 'invariant'.

Lines 136-138: Why are those genes exceptions? And how were exceptions decided upon?

Line 147: How is "high homology" defined?

Figure 2/S1: Legend needs citations for MEGA, MAFFT, etc.

Lines 150-151: Can the authors provide support for this claim beyond just the assembly data?

Were these genes also not observed in the expressed TCR repertoire?

Line 155: This claim should be tempered. Fair to say, no clear orthologue was observed in the assemblies you've analyzed, but not extrapolating to the species as a whole might be a leap.

Lines 156-162: This seems out of place given the header of the results section. Its relevance should be better motivated, perhaps written into its own section.

Line 164: At the beginning of this section, the authors should state which genes from the annotations in Figure 1 are supported at nucleotide level from TCR transcript reads from the animals analyzed.

Line 172 (Figure 3B): Why did the authors combine adult blood and spleen for this analysis?

Line 173: Believe it should be "tendency".

Lines 166-168: The number of animals studied, and their origins should be stated here in text (rather than make reader go methods or figure legends). Are they unrelated animals, littermates, etc.?

Figure 3B: What is the basis for including the genes that are plotted. Usage levels? Is the reader to assume that all other genes have no usage? It would also be useful to comment on the fact that 3 of the 4 genes shown are annotated as ORF, rather than functional. What is the basis for that?

Line 167: What is the rationale for studying these tissues?

Line 177-179: The authors should dissect this claim. What is it based on? If the authors have simply run a statistical test in each group separately and then made this assessment based on individual P values, may not be appropriate. Authors should be more explicit.

Line 190: Metric used for "sharing" (F value) should be stated here in results. "Repertoire sharing" is vague.

Figure 4 legend: typo? "TRG5-3"

Figure 5D: Could the authors comment on low detection of TRGV3-1 in cluster 4? It seems that the frequency of detected gamma TCRs is low relative to delta? The inverse seems to be somewhat true for clusters 3 and 5. How have the authors verified they haven't artificially missed some unidentified TCR chains?

Line 297: number of animals should be stated.

Line 299: Figure 6B, no x axis labels. Looks as though it should be clusters from panel A.

Lines 325: number of animals included in analysis should be stated.

Figure S6C: Potential typo. Should it be "other" rather than "ather"?

Line 427: Typo. "TRCG"

Line 459: The authors should better outline the source of the scaffold assemblies used so it is clear to the reader. Specifically, what technology was used for sequencing, and what methods were used for assembly, as these have consequences on the quality and reliability of annotations. In general, the use of the short-read assemblies for the annotation of TR regions is concerning, and such issues have been consistently raised in the literature. The authors should comment on their choice to use short-read assemblies specifically, rather than other available datasets for naked mole rat that include long-read sequencing data as well. It is particularly concerning that one of the assemblies used is derived from DNA isolated from >1 animal. How the authors have dealt with the presence of heterozygosity in these assemblies should be outlined and discussed. It seems one key aspect of this manuscript is to promote the use of these data as a resource, users need to be made aware of what lengths were taken to ensure the best possible dataset was used and is being provided for the analysis conducted in this study and future studies wishing to use this resource.

Line 470: The authors list many scaffolds that are used for their analysis, but not detail is provided that explains/outlines how these were used to arrive at the data presented in Figure 1. For example the two scaffolds listed for TRG are not identical to one another. How did the authors choose which assembly to use? Did they assess whether differences reflecting sequencing, assembly errors, genuine polymorphism etc.?

Lines 483-484: Details on this should be reported in results. How many genes harbored multiple alleles, and how was the "same position" determined given that the authors are using scaffold, rather than full-locus assemblies?

Reviewer #2 (expert in $\gamma\delta$ T cells):

The naked mole rat (NMR) is a eusocial, subterranean rodent that is increasingly attracting the attention of the biomedical research community. Of interest to this work are its longevity, reduced incidence of tumors and the absence of age-associated thymus involution. The recently reported lack of NK cells is of particular interest, given the generally assumed importance of these cells for tumor control.

Sanchez Sanchez et al used databases and additionally collected data to analyze the (single) transcriptomes of "innate" T cells that could have taken over the function of NK cells. They were able to detect only a very small MAIT cell population and a lack of iNKT cells and discovered a new NK cell-like $\gamma\delta$ T cell population. This population shares an invariant g-chain and a limited TCR δ repertoire with V(D)J genes sharing short homology repeats (microhomology domains) typical for invariant $\gamma\delta$ TCR. The CDR3 of the NMR invariant is identical in length and shows physicochemical similarities with those g-chains of fetal human GV8 positive human fetal gd T cells and fetus-derived mouse (V γ 5/6) $\gamma\delta$ T cells, despite differential GV usage. In contrast to the above-mentioned invariant $\gamma\delta$ T cells, the NMR invariant $\gamma\delta$ T cells are also produced in the adult thymus. Another $\gamma\delta$ T cell population but with a variable TCR repertoire was also identified and this shows a gene expression profile associated with type 3 immune response. In addition, the work contains other interesting information: e.g. annotation of TCR(TR) loci, interspecies comparison of butyrophilin family members, analysis of the $\alpha\beta$ TCR repertoire and (re)definition of T cell

populations based on gene expression patterns.

I think the finding that $\gamma\delta$ T cells can be used as a substitute for a complete population of innate lymphocytes (NK cells) is an interesting and truly new one. NMR is not an "ordinary" model organism and it is (almost) impossible to collect functional data on the immune response. However, this and other studies in recent years show that single-cell transcriptomics (and certainly other single cell-omics in the future) allow by direct comparison of animals with a peculiar life style or phylogenetic origin to define critical features of immune system development and function. Basically, the paper is clearly structured and well written. However, parts of it, the genome analysis in particular, are difficult to understand for non-specialists. In addition, there are points that should be clarified. Here are my detailed comments.

The first section contains the NMR TCR (TR) gene annotation, which is essential for the study. However, Figure 1 is a copy-paste of images from the IMGT website, which was already published in 2021 (<https://www.imgt.org/IMGTrepertoire/LocusGenes/> see the respective Locus representations). It would be appropriate to point this out, especially since the images on the website are better readable and since much more critical information is provided in the links. Instead using these figures for Figure 1, I would recommend creating a new image that contains information that is particularly important for this work, which deals (almost exclusively) with TRG and TRD, and to highlight the relevant genes. It should also be emphasized that the distinction between "functional" on the one hand and "ORF" and "pseudogene" on the other is of very limited significance for statements about functionality.

A surprising finding is the use of the pseudogene TRGC1 by GV3-1. Please briefly describe exactly how this was tested. Was there a sequencing of entire TCR α chain transcripts or is this statement based on that parts of the genomic sequence were found in the transcripts? Please also describe which J and C genes are used by the GV3-2 TCRs.

Figure 2A. I'm not sure if I understood this figure and lines 141-155.

My understanding of the figure is: NMR GV1 branches out with the mouse GV5 (the DETC GV) and further away with the human GV1 containing the invariant GV8. Mouse GV4 stands on its own (I can't read the gene name). NMR GV2 (group) includes human GV9 and NMR GV2. The GV3 group of the mouse (GV5 gene) stands on its own. NMR-GV3 group and genes branch out with the human GV10 gene. The TGV2 group of mice (TRGV4 gene?) branches out with the human or with the NMR-GV4 group (human TRGV10 gene). GV4 of the NMR stands on its own (TRGV4:01,:02,03), as is the case with the mouse GV1 (TRGV1:01, 2:01, 3:01). In any case, please rewrite the relevant text and provide better legend and a better to read figure. Also, it took me some time to learn that the (hard-to-read gene annotation reads from left to right and not from the center outwards (group/gene/gene species). Please also indicate what kind of sequences were compared. V-exon only, protein or nucleotides?

In Figure 1S3. What type of sequences were analyzed ? Nucleotide sequence of genes, ORFs or protein sequences ? In the case of protein sequences, the whole protein or only protein domains. Especially with the BTN3 family, this can lead to different results. In addition, I doubt that the closest relative of the mouse's Skint1 is BTNL1. Mouse Skint1 is a member of a gene family and has homologues (SKINT1L) in many mammalian species. Macaques, for example, have a functional gene. See also: Origin and evolution of dendritic epidermal T cells. Sutoh Y et al. Front Immunol. 2018. The SKINT1-like gene is inactivated in hominoids, but not in all primate species: implications for the origin of dendritic epidermal T cells. Mohamed RH et al. PLoS One. 2015.

Fig. 5. Four abTCR-dominated clusters (0-3), including one CD8ab cluster (3) and two gd-T cell clusters (4 and 5) were identified. Cluster 4 genes are typical of type 3 T cells (curiously without CD4). Cluster 5 expresses CTL/NK-defining genes (Fig. 5C). When reviewing the xls files, I found not information on cytokines. Does this have a physiological significance or is it due to technical problems? Please explain this briefly. In any case, if there is no lymphokine signal (at least not for lymphokines abbreviated with IL-) mentioning that IL-17 is not expressed in cluster 4 is irrelevant. FoxP3, is shown in Fig. 6 as being expressed in individual cells, but not listed corresponding table.

Is this due to the low frequency of FoxP3-positive cells? If yes, it might help to display in Fig. 5 also cytokines in the single-cell plot (similar as in Fig. 6 for FoxP3). A low frequency of cytokine producing cells would be expected under steady state conditions, while surface molecules e.g. CD8b are expressed constitutively by all MHC I restricted cells.

RESPONSE TO REVIEWERS' COMMENTS

Reviewer #1 (expert in annotation and assembly of antigen receptor loci)

(original comment reviewer) Line 98: The legend for Figure 1 does not sufficiently explain features of the figure. Much more detail is needed. In the figures/maps themselves, it is not clear what scaffolds are being used for which parts of the maps, yet the authors say that more than one scaffold is used. And the methods claims that two different assemblies are used even, but there is no indication here what assemblies are represented in these maps. What do the double slanted lines represent. Are these "gaps" in the assembly/scaffolds? What are the black arrows under assembly in panel B? Why are the genes above those arrows named differently? Are they TRA genes? (original reply) We agree with the reviewer that more details can be provided. Importantly, following the main comment of the reviewer, we have revised completely Figure 1. We refer to reviewer #1, points 30-32 for further details.

>>> This legend is still lacking detail. For example, what are the large blue arrows? What are the black arrows? We have explained previously in the legend of Figure 1 the use of the large blue arrows and of the black arrows:

‘Figure 1. Locus representation of the *Heterocephalus glaber* TR loci of the assembly Naked mole-rat maternal (GCA_944319715.1). The figures display the position, nomenclature and functionality of all TR genes according to the IMGT nomenclature⁶⁸. A dotted line ... indicates the distance in kb between the locus and its IMGT bornes²⁸ (5' borne and the most 5' gene in the locus and/or 3' borne and the most 3' gene in the locus) on the top and bottom lines, respectively. These distances are not represented at scale and are not included in the numbers displayed at the right ends of these two lines. Finally, **the black arrows indicate an inverse transcriptional orientation in the locus while the blue arrows indicate the position in the TRG and TRD locus of V genes that are highlighted in the current study.**’

(original comment reviewer) Line 127: The authors mention that some genes have "provisional" names. It's not clear what is mean by this, and based/on compared to what. In addition, this is confusing as the maps shown in Figure 1 look to be claiming that all names are "provisional". Not clear what this means, how it was decided upon, or how to interpret.

(original reply) Because of the use of the chromosomal assemblies (following the important suggestion of this reviewer, see also reviewer #1 points 30-32), there is no provisional nomenclature anymore. The naked mole-rat TR loci have now their definitive nomenclature and NCBI accession numbers have been obtained.

>>> Have the authors corresponded with the TR and IG nomenclature committee of the International Union of Immunological Societies (<https://iuis.org/committees/nom/immunoglobulins-ig-t-cell-receptors-tr-and-major-histocompatibility-mh-nomenclature-sub-committee/>)

This is encouraged if the authors are to claim that the names are definitive and not provisional. This is particularly true given the use of new methodology for genome assembly.

Veronique Giudicelli and Sofia Kossida (author on the current paper) from IMGT® are members of the T-cell Receptor (TR) and Immunoglobulin (IG) Nomenclature Sub-Committee of the IUIS, as evident from the following link: <https://iuis.org/committees/nom/immunoglobulins-ig-t-cell-receptors-tr-and-major-histocompatibility-mh-nomenclature-sub-committee/>. The application for the approval of novel TR nomenclature is hampered by the lack of an official application procedure put forward by this committee.

However, to address the concerns by the reviewer, we would like to highlight that:

- (i) IMGT® is overseen by the University of Montpellier and the CNRS (National Center of Scientific Research) in France. An esteemed Scientific Advisory Board comprising experts in the field is established to provide guidance to IMGT®

- (ii) The data generated by IMGT® are accessible to the global scientific community through the IMGT® portal at <https://www.imgt.org/>
- (iii) IMGT® biocurated data receive accession numbers from the International Nucleotide Sequence Database Collaboration at <https://www.insdc.org/>
- (iv) The IMGT® management system is ISO 9001:2015 certified as seen: https://www.imgt.org/PDF/Certificate_ISO9001_2022_2025.pdf

Taken all together, we did not contact the IUIS committee as there is no procedure yet and we put a series of procedures in place that assures the quality of the TR nomenclature.

(original comment reviewer) Line 164: At the beginning of this section, the authors should state which genes from the annotations in Figure 1 are supported at nucleotide level from TCR transcript reads from the animals analyzed.

(original reply) Given that the primary focus of our article is the $\gamma\delta$ T cell compartment, we have incorporated a detailed table (see also our answer to point 12 of reviewer #1) in the revised version of the manuscript which classifies reads based on the TRGV and TRDV genes they contain (Supplementary Table 2, also provided below) thus allowing a comparison at the level of TCR transcription with the genes depicted in the loci of Figure 1. Regarding TRAV and TRBV usage, we refer now more explicitly to Figure S2 (original submission) where boxplots are provided of TRAV and TRBV usage from the distinct repertoires of neonatal blood samples and adult blood/spleen samples.

(original comment reviewer) Line 172 (Figure 3B): Why did the authors combine adult blood and spleen for this analysis?

(original reply) We aimed to highlight the preferential usage of specific TRDV genes in neonates compared to other stages. To enhance the statistical power and increase our sample size (n), we merged the two adult datasets. We would like to perform new bulk TCR-seq experiments to increase n in the different age and tissue groups.

>>> As long as this is clearly stated rationale in the manuscript, seems reasonable to me. We now state in the legend of (new) Fig. 4B: ‘The adult blood and spleen data sets were combined to increase statistical power.’

(original comment reviewer) Line 167: What is the rationale for studying these tissues?

(original reply) The primary factor motivating the choice of spleen and blood for this first investigation of the TCR repertoire in the naked mole-rat is the feasibility of obtaining these tissues at various developmental timepoints, including neonates. The naked mole-rat represents an unconventional animal model, and the procedures for collecting organs and optimizing these collection methods have not been comprehensively documented. We think that investigating immune cells associated with other tissues than spleen and blood in the naked mole-rat is of interest and should be a topic in future studies but is out of scope of the current study.

>>> As long as this is clearly stated rationale in the manuscript, seems reasonable to me.

We have included now in the manuscript (Methods section, page 19):

‘The primary factor motivating the choice of spleen and blood for investigation of the TCR repertoire in peripheral tissues was the feasibility of obtaining these tissues in the naked mole-rat at various developmental timepoints, including neonates. The cervical and thoracic thymy were included in the study as they are the T cell-generating organs in the naked mole-rat.’

(original comment reviewer) 21. Line 177-179: The authors should dissect this claim. What is it based on? If the authors have simply run a statistical test in each group separately and then made this assessment based on individual P values, may not be appropriate. Authors should be more explicit.

(original reply) Is regarding: ‘Differences in junctional diversity, CDR3 nucleotide length and NDN size between neonates and adults were more prominent in TRDV1-6-containing sequences than TRDV1-4 (Fig. 3C).’

As highlighted by the reviewer, the p-values in the figure correspond to comparisons that are not directly connected to the statement. Furthermore, the presence of non-significant differences in some of these comparisons could potentially lead to confusion for the reader. In this last case we think that this is explained by the low number of samples compared, and the statistical test used (Kruskal-Wallis test). Because the current p values depicted in Fig. 3C (original submitted version; now Fig .4C), we removed these and we adapted the

corresponding text in the revised manuscript (Results section, page 8). We would like to increase *n* in the different groups to enhance the statistical power.

(original comment reviewer) 24. Figure 5D: Could the authors comment on low detection of TRGV3-1 in cluster 4? It seems that the frequency of detected gamma TCRs is low relative to delta? The inverse seems to be somewhat true for clusters 3 and 5. How have the authors verified they haven't artificially missed some unidentified TCR chains?

(original reply) The reason for the observed differences in TRG and TRD detection between Clusters 4 and 5 can be related to the fact the different types of $\gamma\delta$ T cells can express different $\gamma\delta$ TCR levels. In the mouse, the $V\gamma5+$ $\gamma\delta$ subset has been shown to express high TCR levels compared to other $\gamma\delta$ subsets (e.g., Sumaria 2011 J Exp Med doi/10.1084/jem.20101824; Van hede 2017 PNAS doi/10.1073/pnas.1712883114), while in human the $V\delta1$ subset express higher TCR levels than the $V\delta2$ subset (e.g., Yokobori et al 2009 Clin Exp Immunol 10.1111/j.1365-2249.2009.03974.x; Jagannathan et al 2014 Sci Transl Med 0.1126/scitranslmed.3009793). We therefore propose that the two $\gamma\delta$ T cell clusters detected in the naked mole-rat, that are associated with different effector profiles, show also such differences in TCR expression levels which is then reflected in different levels of TRG and TRD CDR3 detection by the TRUST4 package based on single-cell gene expression.

The transcriptomic profile of Cluster 3 (as depicted in Figure 5C, original submission) and the TCR sequences identified by TRUST4 (as illustrated in Figure 5B, original submission) suggests that Cluster 3 predominantly comprises CD8 TCR $\alpha\beta$ cells. Despite this conclusion, as the reviewer pointed out, we are able to detect CDR3 γ sequences in this cluster. This seemingly contradictory result can be explained by the fact that TCR $\alpha\beta$ T cells have the capacity to express transcripts of rearranged TRG sequences (Sherwood et al. 2011 Sci Transl Med 10.1126/scitranslmed.3002536). In line with this, almost no CDR3 δ sequences can be detected in this TCR $\alpha\beta$ cluster, a result that is expected as TRD genes are interspersed with TRA genes in the same loci (as illustrated in Figure 1): the expression of a TRA chain excludes the expression of a TRD chain.

>>> This interpretation should be included in the manuscript, in my opinion.

Following the suggestion of the reviewer, we have now included this interpretation in the Results section of the revised manuscript (page 11).

(original comment reviewer) 30. Line 459: The authors should better outline the source of the scaffold assemblies used so it is clear to the reader. Specifically, what technology was used for sequencing, and what methods were used for assembly, as these have consequences on the quality and reliability of annotations. In general, the use of the short-read assemblies for the annotation of TR regions is concerning, and such issues have been consistently raised in the literature. The authors should comment on their choice to use short-read assemblies specifically, rather than other available datasets for naked mole rat that include long-read sequencing data as well. It is particularly concerning that one of the assemblies used is derived from DNA isolated from >1 animal. How the authors have dealt with the presence of heterozygosity in these assemblies should be outlined and discussed. It seems one key aspect of this manuscript is to promote the use of these data as a resource, users need to be made aware of what lengths were taken to ensure the best possible dataset was used and is being provided for the analysis conducted in this study and future studies wishing to use this resource.

(original reply) In our prior work, we relied on scaffold-based assemblies generated from BGISEQ-500 sequencing, specifically the Heter_glaber.v1.7_hic_pac62 and the HetGla_female_1.063 assemblies, as they were the sole available assemblies in 2021 on NCBI for the naked mole-rat. These assemblies served as an expedient means for the temporary annotation of the TR loci in the naked mole-rat, as well as for subsequent analyses of TR expression data. Importantly, we have now completed the full annotation of all four TR loci using the latest complete chromosome haploid assemblies, which were derived from long-read sequencing, aligning with the request of the reviewer. This accomplishment has led to the establishment of the definitive nomenclature for the naked mole-rat TR loci which is now presented in Figure 1 of the revised manuscript. This Figure 1 is also provided below.

Accession numbers have been obtained at the NCBI:

TRAD locus maternal BK064754

TRG locus paternal BK064755

TRG locus maternal BK064756

TRB locus paternal BK064758

TRB locus maternal BK064759

Notably, this final updated annotation has not resulted in substantial changes to the provisional nomenclature. Consequently, all analyses related to TR expression data in the naked mole-rat remain valid and have been further confirmed.

>>> The additional annotation of updated assemblies for this species is noted, and the effort is appreciated by the reviewer.

a) It should still be stated for the reader what the source of these assemblies is, other than just the accession. It is useful for a reader to understand how assemblies were arrived at; this gives the reader an understanding of what the quality of the assemblies is, etc.

Following this comment, we have now included in the Methods section (page 17) of the revised manuscript the following text:

‘Two chromosome-level *Heterocephalus glaber* genome assemblies, the naked mole-rat maternal (GCA_944319715.1) and paternal (GCA_944319725.1), were obtained from the National Library of Medicine (NCBI) Assembly database to study the TR loci. Maternal and paternal assemblies were derived from the gonad tissue of a 5-year-old male naked mole-rat generated via the 10x Linked reads sequencing and Nanopore long-read sequencing respectively.’

b) Detail on the use of maternal and paternal assemblies should be provided in the results section. The authors make no mention of this, yet Figure 1 includes a statement about the map shown coming from a “maternal” assembly. This will be confusing for readers if not explained.

We adjusted the Results section (page 5) based on this comment:

‘To determine the type of expressed naked mole-rat TCRs by high-throughput sequencing, we first annotated the TR loci (TRG, TRAD and TRB) of *Heterocephalus glaber* (Fig. 1). While annotations were conducted for the TRG and TRB loci in both the maternal and paternal chromosomal assemblies, the annotation for the TRAD locus was exclusively carried out in the maternal assembly. This decision was prompted by the presence of gaps in the paternal assembly of the TRAD locus, indicative of low-quality sequencing in that genome region. Of note, there were no differences observed in the presence or order of TR genes between the annotated maternal and paternal haplotypes for TRB and TRG loci.’

c) With respect to differences in annotations, the explanations below are somewhat hard to follow. The authors note some of the differences with their previous annotations from scaffolds, but seemingly not all? For example, TRAV12-5 is not mentioned (and this gene doesn’t appear to sit adjacent to a “gap”)? Is that gene being removed from the overall annotation set? What were the final decisions made to deal with discrepancies between the previous scaffold and current annotations. Are all being combined into one database for analysis in this paper?

We thank the reviewer of this comment. Indeed TRAV12-5 was not found in the new assemblies and was accidentally missing from the previous list. However, this gene does appear next to a gap.

As mentioned earlier, the core TR gene sets demonstrate consistency across scaffolds and the updated assemblies, with minimal variations. Simultaneously, the new assemblies exhibit significantly improved quality, devoid of any gaps. Consequently, in this paper, we opt to showcase annotations derived from the fresh analysis of chromosomal assemblies, exclusively featuring genes identified within the chromosome sequences.

Currently, all genes of a species are consolidated into the reference set, which is utilized in IMGT/HighV-QUEST. IMGT® will release the 'user-customizable germline sets' to the broader scientific community through IMGT/HighV-QUEST before the end of 2024.

d) The numbers of genes/alleles annotated and potential allelic variants should be stated in results section. Given that the authors have annotated a maternal and paternal chromosome from the species, this should be presented clearly in the results so the reader understands what data is being presented (e.g., in Figure 1)

We followed the request of the reviewer, and we enriched the Results section (page 5) with supplementary gene information on the maternal and paternal assembly:

‘All the genes were present in both maternal and paternal assemblies and polymorphisms between the two assemblies were detected in 6 genes: 3 TRGV genes (TRGV1-3, TRGV1-4 and TRGV4), 1 TRGJ gene (TRGJ3-2) and 2 TRGC genes (TRGC2 and TRGC5)...

The genes were present in both maternal and paternal assemblies, while polymorphic variations were observed in 5 genes across the two assemblies: 2 TRBV genes (TRBV9 and TRBV27), 1 TRBJ gene (TRBJ1-6) and 2 TRBC genes (TRBC1 and TRBC2).’

e) What efforts have been made to assess the accuracy of assemblies spanning coding genes? This should be stated.

To assess the precision of assemblies encompassing coding genes, the initial step involved revising the estimated count of V, D, J, and C genes at the locus, comparing it with the closest species from an evolutionary perspective. Subsequently, the examination included the assessment of the number and length of gaps, as well as the identification of vectors. Additionally, a thorough analysis of gene organization was conducted, considering data from species most closely related to the species of interest. Finally, the detection of IMGT boundaries (IMGT bornes) was explored as it could offer further confirmation of the assembly's quality. We have incorporated this information into the results section of the revised manuscript (page 5):

‘The evaluation of the accuracy of the assemblies was performed through the detailed revision of the estimated number of variable, diversity, joining and constant genes at the locus, along with their organization, in comparison with the data of the most closely related evolutionary species.’

f) In the TPA submissions, why was there no TRA/D submitted for the paternal chromosome? Of note, you mention a comparison of maternal and paternal assemblies for the TRB and TRG loci, but nothing is mentioned for TRA/D.

Annotations were conducted for the TRG and TRB loci in both chromosomal assemblies. For the TRAD locus, annotation was performed exclusively in the maternal assembly, due to the presence of gaps in the paternal assembly indicating lower quality. This explanation has been added to the revised manuscript:

Results section (page 5):

‘...the annotation for the TRAD locus was exclusively carried out in the maternal assembly. This decision was prompted by the presence of gaps in the paternal assembly of the TRAD locus, indicative of lower quality sequencing in that genome region.’

Methods section (page 17):

‘Note that we did not submit a naked mole-rat annotation for the TRAD locus based on the paternal assembly because of the presence of gaps, indicative of lower quality sequencing in that genome region.’

g) The authors state that TRB and TRG did not show differences in gene content, but were there polymorphisms in genes? Were these catalogued as alleles in your dataset? This needs to be stated in the results.

We followed the request of the reviewer, and included in the Results section (page 5) supplementary gene information on the maternal and paternal assembly:

‘All the genes were present in both maternal and paternal assemblies and polymorphisms between the two assemblies were detected in 6 genes: 3 TRGV genes (TRGV1-3, TRGV1-4 and TRGVD), 1 TRGJ gene (TRGJ3-2) and 2 TRGC genes (TRGC2 and TRGC5)....’

The genes were present in both maternal and paternal assemblies, while polymorphic variations were observed in 5 genes across the two assemblies: 2 TRBV genes (TRBV9 and TRBV27), 1 TRBJ gene (TRBJ1-6) and 2 TRBC genes (TRBC1 and TRBC2).’

Additional comments:

1) Changes to methods section pertaining to phylogenetic analysis are noted. In this section, more detail should be provided with respect to the parameters used for MAFFT and ML tree estimation.

We have extended the Methods section (pages 18-19 of the revised manuscript) according to the reviewer’s comment, incorporating software tools parameters and software versions.

2) For TCR repertoire analysis, what germline gene annotations were used. Were those analyses updated to reflect the new annotations? Were only the long-read assemblies used given the issues/gaps in the previously used scaffolds? This needs to be clearly stated.

We have used the gene annotations based on the long-read assemblies. This is now stated in the revised paper (Methods section, page 20).

Reviewer #2 (expert in $\gamma\delta$ T cells)

The authors have responded carefully and to my satisfaction to my criticism. The errors have been corrected and both text and illustrations have been changed accordingly. The manuscript is now easier to understand and the hypotheses and interpretations put forward are sufficiently justified.

We thank the reviewer for their previous comments allowing to improve our manuscript and we are happy to see that the reviewer is satisfied with the changes.

Reviewer #3 (expert in the naked mole-rat)

On the basis of the phenomenon that the naked mole-rat lack the natural killer cell population the potent weaponry against cancer cell development the authors provided comprehensive evidences for the existence of an alternative immune system strategy in this unique species serving as main arm for immunosurveillance and anticancer response. The formation of large portions of gammadelta T-cells produced by both the thoracic and cortical thymus until adult life underlines the existence of this alternative immunosurveillance mechanism in naked mole-rats.

These new findings help to elucidate some of relevant questions in regards to the unexpected longevity of this species and its general cancer-resistance. The applied methodology is sound and provides new data about the function of the immune system in naked mole-rat. This publication will add substantial new information about the unique physiology of this subterranean-living rodent.

We sincerely thank the reviewer for this to-the-point summary of our paper and the appreciation of our work.

REVIEWER COMMENTS

Reviewer #1:

[Note from editor: This reviewer was no longer available to provide comments to the authors and was therefore replaced by Reviewer #4.]

Reviewer #4 (Remarks to the Author):

I have been asked to comment specifically on the suitability of the nomenclature proposed by IMGT for what is being proposed as a reference assembly, resource and will likely set the baseline for future studies.

It is clear that IMGT has been a critical community resource for many years, open and underpinned so many studies. However, it is not an official repository, it is only as useful as the community engages. Having spent some time looking and reaching out to IMGT Committee members, there is no evidence that any external committee oversight or meeting have ever taken place. IUIS does have a mandate to drive consistency in nomenclature across immune gene loci, and has been building the structure to achieve this. I appreciate that IMGT have been a part of this process, but again, can find no evidence written or verbal from member of the IUIS TR/IG nomenclature committee that they have ever attended a meeting or contributed outside of IMGT specific challenges/aims.

The above is intended to understand the rebuttal to the reviewer, from the perspective of both, pasted below.

Veronique Giudicelli and Sofia Kossida (author on the current paper) from IMGT® are members of the T-cell Receptor (TR) and Immunoglobulin (IG) Nomenclature Sub-Committee of the IUIS, as evident from the following link: <https://iuis.org/committees/nom/immunoglobulins-ig-t-cell-receptors-tr-and-majorhistocompatibility-mh-nomenclature-sub-committee/>. The application for the approval of novel TR nomenclature is hampered by the lack of an official application procedure put forward by this committee.

However, to address the concerns by the reviewer, we would like to highlight that: (i) IMGT® is overseen by the University of Montpellier and the CNRS (National Center of Scientific Research) in France. An esteemed Scientific Advisory Board comprising experts in the field is established to provide guidance to IMGT®

(ii) The data generated by IMGT® are accessible to the global scientific community through the IMGT® portal at <https://www.imgt.org/>

(iii) IMGT® biocurated data receive accession numbers from the International Nucleotide Sequence Database Collaboration at <https://www.insdc.org/>

(iv) The IMGT® management system is ISO 9001:2015 certified as seen: https://www.imgt.org/PDF/Certificate_ISO9001_2022_2025.pdf

Taken all together, we did not contact the IUIS committee as there is no procedure yet and we put a series of procedures in place that assures the quality of the TR nomenclature.

Although the above is factual to a large extent, having researched both nomenclature groups, it does not rebut the reviewer's comments, and a process of engagement with the IUIS Committee (of which the confirm they are members) was not undertaken.

I do not doubt the validity of the nomenclature proposed in this paper, but do wonder if this system will be adaptable to loci that vary in location or presence/absence polymorphism in future studies. This is precisely why consistency and community agreement is so important for setting the baseline nomenclature. The only compromise I can see is that the manuscript states that the proposed nomenclature is considered provisional based on future data being available and involvement of IUIS at the appropriate time/level.

RESPONSE TO REVIEWERS' COMMENTS

Reviewer #4

I have been asked to comment specifically on the suitability of the nomenclature proposed by IMGT for what is being proposed as a reference assembly, resource and will likely set the baseline for future studies.

It is clear that IMGT has been a critical community resource for many years, open and underpinned so many studies. However, it is not an official repository, it is only as useful as the community engages. Having spent some time looking and reaching out to IMGT Committee members, there is no evidence that any external committee oversight or meeting have ever taken place. IUIS does have a mandate to drive consistency in nomenclature across immune gene loci, and has been building the structure to achieve this. I appreciate that IMGT have been a part of this process, but again, can find no evidence written or verbal from member of the IUIS TR/IG nomenclature committee that they have ever attended a meeting or contributed outside of IMGT specific challenges/aims.

The above is intended to understand the rebuttal to the reviewer, from the perspective of both, pasted below. Veronique Giudicelli and Sofia Kossida (author on the current paper) from IMGT® are members of the T-cell Receptor (TR) and Immunoglobulin (IG) Nomenclature Sub-Committee of the IUIS, as evident from the following link: <https://iuis.org/committees/nom/immunoglobulins-ig-t-cell-receptors-tr-and-majorhistocompatibility-mh-nomenclature-sub-committee/>. The application for the approval of novel TR nomenclature is hampered by the lack of an official application procedure put forward by this committee. However, to address the concerns by the reviewer, we would like to highlight that: (i) IMGT® is overseen by the University of Montpellier and the CNRS (National Center of Scientific Research) in France. An esteemed Scientific Advisory Board comprising experts in the field is established to provide guidance to IMGT® (ii) The data generated by IMGT® are accessible to the global scientific community through the IMGT® portal at <https://www.imgt.org/> (iii) IMGT® biocurated data receive accession numbers from the International Nucleotide Sequence Database Collaboration at <https://www.insdc.org/> (iv) The IMGT® management system is ISO 9001:2015 certified as seen: https://www.imgt.org/PDF/Certificate_ISO9001_2022_2025.pdf Taken all together, we did not contact the IUIS committee as there is no procedure yet and we put a series of procedures in place that assures the quality of the TR nomenclature.

Although the above is factual to a large extent, having researched both nomenclature groups, it does not rebut the reviewer's comments, and a process of engagement with the IUIS Committee (of which the confirm they are members) was not undertaken.

I do not doubt the validity of the nomenclature proposed in this paper, but do wonder if this system will be adaptable to loci that vary in location or presence/absence polymorphism in future studies. This is precisely why consistency and community agreement is so important for setting the baseline nomenclature. The only compromise I can see is that the manuscript states that the proposed nomenclature is considered provisional based on future data being available and involvement of IUIS at the appropriate time/level.

We followed the suggestion of the reviewer and have added the following statement to the manuscript (page 18, highlighted) :

'The proposed nomenclature is considered provisional based on future data being available and involvement of IUIS at the appropriate time/level.'